# Text-Augmented Multimodal LLMs for Chemical Reaction Condition Recommendation

## Abstract

High-throughput reaction condition (RC) screening is fundamental to chemical synthesis. However, current RC screening suffers from laborious and costly trial-and-error workflows. Traditional computer-aided synthesis planning (CASP) tools fail to find suitable RCs due to data sparsity and inadequate reaction representations. Nowadays, large language models (LLMs) are capable of tackling chemistry-related problems, such as molecule design, and chemical logic Q&A tasks. However, LLMs have not yet achieved accurate predictions of chemical reaction conditions. Here, we present Chemma-RC, a text-augmented multimodal LLM that responds to task-specific questions by generating answers about reaction conditions. It learns a unified reaction representation via modality alignment from a corpus of reactions and question prompts, molecular structures in SMILES format, and graphical representations of chemical reactions. We construct a 1.2 million pair-wised Q&A instruction dataset to train Chemma-RC and design a projection module for modality alignment. Our experimental results demonstrate that Chemma-RC achieves state-of-the-art performance on two open benchmark datasets and exhibits strong generalization capabilities on out-of-domain (OOD) and High-Throughput Experimentation (HTE) datasets. Chemma-RC has the potential to accelerate high-throughput condition screening in chemical synthesis.

## 1 Introduction

Chemical synthesis is a crucial step for the discovery of transformative molecules in multiple fields, including drug design, materials, renewable energy, etc. In chemical synthesis, reaction conditions are usually optimized to maximize the yield of each target molecule or minimize the cost of the corresponding process (Shields et al., 2021; Taylor et al., 2023). Despite significant advancements in chemical synthesis over the past few decades, discovering suitable reaction conditions from the extensive substrates combined with high-dimensional conditions renders exhaustive experimental impractical (Angello et al., 2022). Chemists have focused on building reliable and convenient computer-aided synthesis planning (CASP) tools to facilitate chemical synthesis (Corey & Wipke, 1969; Mikulak-Klucznik et al., 2020; Schwaller et al., 2021). However, few efforts have been made to solve the problem of reaction condition screening due to the low sparsity of chemical data, and the lack of effective reaction representation (Mehr et al., 2020; Rohrbach et al., 2022). *In summary, to realize efficient synthesis in chemistry, there is an urgent need to realize high-efficiency reaction condition recommendations.*

There are various types of data in the field of chemistry, including simplified molecular-input line-entry system (SMILES) (Weininger et al., 1989), graphs, and textual corpus of reaction (Schlichtkrull et al., 2018), which encompasses the descriptions of reaction processes, and reaction mechanisms. Traditional methods tackling the reaction condition recommendation (RCR) task typically rely on sequence-based SMILES data for end-to-end training (Gao et al., 2018; Schwaller et al., 2019; Andronov et al., 2023). However, training exclusively on sequence-based SMILES representations may hinder the model's ability to capture the difference between similar reactions, as the feature distances encoded by transformers may be too close in the representation space. The capability to encode different reactions is critical for prediction, as even minor variations in a substrate's functional group can result in fundamentally different reaction conditions. Therefore, it is necessary to incorporate additional information into reaction representations for RCR tasks. Given that the tex-

tual corpus contains chemical knowledge, which is invaluable for a comprehensive understanding of reactions, we aim to leverage cross-modality data to predict reaction conditions precisely.

Nowadays, the emergence of generative large language models (LLMs), typified by GPT-4, has sparked significant interest in the field of AI for chemistry (Baum et al., 2021; Achiam et al., 2023; Boiko et al., 2023; Guo et al., 2023; M. Bran et al., 2024). Large multimodal models (LMMs) have demonstrated remarkable predictive capabilities in integrating modalities such as vision, text, and speech (Li et al., 2023; Zhu et al., 2024; Liu et al., 2024a). Therefore, we hypothesize that LMMs endowed with LLMs' foundational capabilities in chemistry can deal with various modalities of chemical data, thereby enhancing the predictive performance in chemical tasks. However, it presents a significant challenge in designing modules to integrate various modalities effectively. Hence, *it is imperative to develop an effective prediction model that can incorporate different chemical data into LLMs to achieve a more comprehensive understanding of reaction processes, facilitating the task of chemical reaction condition recommendation.*

In view that molecules can be expressed as sequences, and reactions are described as natural language, e.g. text corpus, LMMs can be a potential solution due to the following advantages: (i) foundational LLMs can learn relationships between molecules in reactions, thereby acquiring chemical knowledge akin to the learning process of chemists (Achiam et al., 2023); (ii) via learning the joint representation of chemical reactions from different modalities, including graphs, SMILES, and corpus, LLMs might be empowered to understand the mechanism of reactions, which facilitates the task of RCR. To this end, we fine-tune general-purpose LLMs with domain-specific reaction data. Specifically, we present Chemma-RC, a multimodal LLM that jointly learns from the SMILES, graphs, and textual corpus of reactions. The contributions of this work can be summarized as follows:

1. We propose a multimodal LLM, a.k.a. Chemma-RC, to jointly learn representation from SMILES, graphs, and textual corpus of reactions for condition recommendation tasks. We further develop two distinct types of condition prediction modules, a classification module, and a generation module for Chemma-RC to enhance its compatibility with different reaction condition combinations.

2. We design text-augmented instruction prompts to construct a 1.2 million pair-wised Q&A dataset for training. We propose the Perceiver (Jaegle et al., 2021) module for modality alignment, which utilizes latent queries to align graphs and SMILES tokens with text-related tokens.

3. Through experimental validation on benchmark datasets, Chemma-RC achieves competitive results comparable to state-of-the-art models. Furthermore, Chemma-RC exhibits strong generalization capabilities on out-of-domain (OOD) and high-throughput experimentation (HTE) datasets.

## 2 RELATED WORK

In chemical synthesis, reaction conditions are usually developed and optimized to maximize the yield of each target molecule or minimize the cost of the corresponding process (Shields et al., 2021; Taylor et al., 2023). High-throughput reaction condition (RC) screening, as an important tool in synthesizing molecules, exerts an important influence on chemical synthesis. However, discovering suitable reaction conditions from the extensive matrix of substrates combined with the high-dimensional reaction conditions renders exhaustive experimental impractical. (Angello et al., 2022). For decades, chemists have focused on building reliable and convenient computer-aided synthesis planning (CASP) tools to facilitate chemical synthesis (Corey & Wipke, 1969; Mikulak-Klucznik et al., 2020). For instance, Coley et al. built a multiway classification model based on a two-step graph convolutional network (GCN) for the reaction prediction task (Coley et al., 2017; 2019). Due to the effectiveness of a simplified molecular-input line-entry system (SMILES) (Weininger et al., 1989), as strings of a context-free, Nam et al. proposed the first sequence-to-sequence model for forward prediction using the SMILES representations of molecules (Nam & Kim, 2016). Inspired by attention-based transformer model (Vaswani et al., 2017), Schwaller et al. proposed molecular transformers (Schwaller et al., 2019; Ding et al., 2024), which were applied in forward prediction and reaction condition recommendation (RCR) tasks (Schwaller et al., 2019; Andronov et al., 2023).

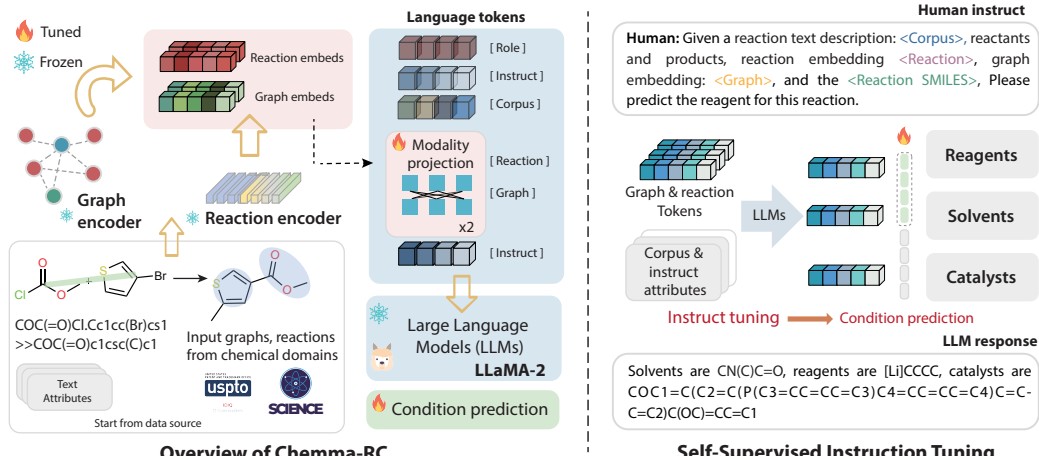

Figure 1: **Architecture of Chemma-RC.** Chemma-RC processes task-specific questions constructed by text-augmented multimodal instruction prompts and generates answers. Specifically, it takes three modalities of data as inputs: text (a textual corpus of reactions and question prompts), molecular SMILES, and reaction graphs. Two distinct types of prediction modules, a classification module, and a generation module are proposed to predict chemical reaction conditions.

Chemical reaction condition recommendation tasks aim to recommend catalysts, reagents, solvents, or other conditions for a specific reaction. The exploration of a suitable condition is crucial for the realization of CASP, as it dictates the expected outcomes, including reaction yields and rates (Schnitzer et al., 2024). Gao et al. developed a neural network model to predict the chemical context as well as the temperature for any particular organic reaction (Gao et al., 2018); Maser et al. proposed a machine-learned ranking model to predict the set of conditions used in a reaction as a binary vector (Maser et al., 2021); Wang et al. proposed Parrot, a powerful and interpretable transformer-based model for the prediction of reaction condition (Wang et al., 2023a); In the meantime, in order to enhance the representation of reactions, Qian et al. (Qian et al., 2023) designed TextReact, which introduced relevant corpus retrieved from literature to enhance the molecular representation of the reaction based on SMILES. Nevertheless, these methods rely on manual feature selection by experts' knowledge and lack a general prediction model with powerful reaction representation.

Nowadays, the emergence of generative pre-trained transformer-based large language models (LLMs), typified by GPT-4, has triggered keen interest in leveraging such techniques to tackle chemistry challenges (Baum et al., 2021; Achiam et al., 2023). Several works focus on chemical agents for the exploration of chemical conditions. Boiko et al. (Boiko et al., 2023) proposed a GPT-4 driven scientific agent system to plan and perform complex experiments, which accelerates reaction condition screening and experimental automation in chemistry; Bran et al. developed ChemCrow, which augmented LLMs with chem-expert-designed tools (M. Bran et al., 2024); However, for tasks demanding a precise understanding of molecular SMILES representation, such as reaction prediction, and retrosynthesis, LLMs exhibited a less competitive performance than traditional machine learning baselines (Guo et al., 2023). Partially, the reason is that, without an in-depth understanding of the SMILES strings, and the reaction process that transforms reactants into products, it will be difficult for LLMs to generate accurate responses.

Besides SMILES strings, there are various types of data such as molecule graphs and the reactions' external textual corpus in the chemistry synthesis field. By synergizing the strengths of multiple modalities, large multimodal models (LMMs) can achieve higher accuracy, and perform more effectively in a wide range of applications (Edwards et al., 2022; Li et al., 2023; Zhu et al., 2024; Liu et al., 2024a; Li et al., 2024; Liu et al., 2024b).

## 3 METHODS

### 3.1 PROBLEM SETUP

For a task of reaction condition recommendation, we define the $X$ as the input for the chemical reaction $R$, $T$ as the reaction corpus, $G$ as the graph representations of reactions, and the output

$Y$ as a list of reaction conditions including the catalyst, solvent, and reagent. Thus, we define the prediction model $\mathcal{F}$, i.e., $Y = \mathcal{F}(X, G, T)$.

In this paper, we incorporate three types of data for the training of model $\mathcal{F}$:

1. **SMILES of a reaction** $X$: each example in the training set is presented by chemical SMILES, i.e., *"CC(C)O.O=C(n1ccnc1)nccnc1 >> CC(C)OC(=O)n1ccnc1"*.

2. **Graphs of a reaction** $G$: each SMILES representation of the reactants and the product is encoded using a graph neural network (GNN). All compounds are integrated to generate a comprehensive reaction representation.

3. **An unlabeled reaction corpus**: a paragraph describing a chemical reaction, e.g., "To a solution of CDI (2 g, 12.33 mmol), in DCM (25 mL) was added isopropyl alcohol (0.95 mL, 12.33 mmol) at 0° C.".

## 3.2 MODEL STRUCTURE

Here we first introduce the **Chemma-RC**, a multimodal LLM designed for reaction condition recommendation (RCR). An overview of Chemma-RC is illustrated in Figure. 1. Chemma-RC responds to task-specific questions constructed by instruction prompts such as *"Please predict the reagent for this reaction."*, and generates answers about reaction conditions. The Chemma-RC model accepts three different data modalities as inputs. This includes text from a corpus of reactions and question prompts, molecular structures in SMILES format, and graphical representations of chemical reactions. We employ both transformer-based reaction encoder and GCN models to learn reaction representations from SMILES and graph structure jointly. Subsequently, the modality projection transforms the graph and SMILES embeddings into language tokens compatible with LLM space. These learnable tokens, defined as graph and reaction tokens, along with tokens of instruction prompts, are then input into the LLM to predict chemical reaction conditions. Note that, we develop two distinct types of condition prediction modules, a **classification** and a **generation** prediction module to enhance its compatibility with different chemical reaction conditions. On the one hand, the reason for performing classification tasks is to select the most suitable reaction conditions from commercially available libraries, as it is common practice to prioritize purchasable molecules. On the other hand, the generation module can assist in designing novel molecules, which can be obtained by synthesis experiments conducted. Therefore, we define two distinct tasks including classification and generation modules to address these objectives. Furthermore, existing baseline methods treat RCR as a classification task for the USPTO-Condition datasets. To ensure a fair comparison, we conduct a classification module for prediction and evaluation.

### 3.2.1 CONSTRUCTION OF TEXT-AUGMENTED INSTRUCTION PROMPTS

Instruction prompt datasets refer to format structured or unstructured data as natural language instructions so that LLMs can respond properly (Reynolds & McDonell, 2021; Wang et al., 2023b). Compared to creating language instruction datasets for fine-tuning LLMs, constructing multimodal instruction datasets requires a thorough understanding of domain-specific tasks. Recent advancements indicate that the other data modalities, such as images, and graphs, can be transformed as the prefix of prompts thereby facilitating effective reasoning based on inputs (Tsimpoukelli et al., 2021; Zhu et al., 2024; Liu et al., 2024a).

Toward reaction condition recommendation task in chemical synthesis, we design a tailored instruction prompt system for better cross-modality alignment and instruction tuning (Figure. 2). Compared to instruction prompts for natural language instruction tuning (Figure. 2(a)), we introduce augmented text tokens and multimodal tokens into instruction prompts (Figure. 2(b)). To be specific, given a reaction, we retrieve a relevant corpus—a paragraph containing contextual information that closely resembles the reaction—and populate the <Corpus>placeholder with this data. Next, the reaction is converted into its corresponding SMILES representation, which is then inserted into the <Reaction SMILES>placeholder. Finally, we introduce two additional placeholders, <Reaction>and <Graph>, designed to accommodate the reaction and graph-based representations, respectively. In instruction fine-tuning, all reaction embedding representations are extracted by reaction encoders. Via the modality alignment module, all embeddings are inserted into token placeholders to align text-related tokens in language space. We also give pseudo-code as follows to explain this integration process, which can be found in the Appendix. C Algorithm 1.

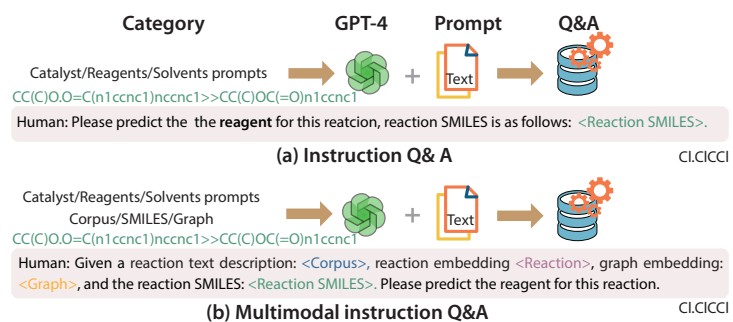

Figure 2: Instruction of text-augmented prompts. **(a)** Traditional instruction prompts for natural language instruction tuning; **(b)** Our proposed text-augmented multimodal instruction Q&A prompts.

### 3.2.2 ENCODER AND DECODER

Given a reaction $R$, we adapt a pioneering transformer-based encoder, Parrot (Wang et al., 2023a) to produce the reaction embeddings $\mathbf{X}_R \in \mathbb{R}^{N \times C}$. Here, $N$ and $C$ indicate the length of text tokens and embedding channels, respectively. During training, the encoder computes a contextual vector representation of the reactions by performing self-attention on the masked canonicalized SMILES string of molecules. We denote reaction embeddings as SMILES embedding in the following section.

In the meantime, we leverage a GNN (Schlichtkrull et al., 2018) to model the relationship between atoms in molecules. We denote directed and labeled multi-graphs as $G = (\mathcal{V}, \mathcal{E}, \mathcal{R})$ with nodes (atom entities), $v_i \in \mathcal{V}$ and labeled edges (atom relations) $(v_i, r, v_j) \in \mathcal{E}$, where $r \in \mathcal{R}$ is a relation type. GNN can be understood as special cases of a simple differentiable message-passing framework:

$$h_i^{(l+1)} = \sigma \left( \sum_{m \in \mathcal{M}_i} g_m \left( h_i^{(l)}, h_j^{(l)} \right) \right) \tag{1}$$

where $h_i^{(l)} \in \mathbb{R}^{d^{(l)}}$ is the hidden state of node $v_i$ in the $l$-th layer of the neural network, with $d^{(l)}$ being the dimensionality of this layer's representations. Incoming messages of the form $g_m(\cdot, \cdot)$ are accumulated and passed through an element-wise activation function $\sigma(\cdot)$, such as the $\mathrm{ReLU}(\cdot) = \max(0, \cdot)$, $\mathcal{M}_i$ denotes the set of incoming messages for node $v_i$ and is often chosen to be identical to the set of incoming edges. $g_m(\cdot, \cdot)$ is typically chosen to be a (message-specific) neural network-like function or simply a linear transformation $g_m(h_i, h_j) = W h_j$ with a weight matrix $W$. Motivated by this architecture, GCNN (Schlichtkrull et al., 2018) proposed a refined propagation model for the forward-pass update of an entity or node:

$$h_i^{(l+1)} = \sigma \left( \sum_{r \in \mathcal{R}} \sum_{j \in \mathcal{N}_i^r} \frac{1}{c_{i,r}} W_r^{(l)} h_j^{(l)} + W_0^{(l)} h_i^{(l)} \right) \tag{2}$$

where $\mathcal{N}_i^r$ denotes the set of neighbor indices of node $i$ under relation $r \in \mathcal{R}$. $c_{i,r}$ is a problem-specific normalization constant that can either be learned or chosen in advance (such as $c_{i,r} = |\mathcal{N}_i^r|$).

We develop two distinct types of prediction modules, a classification module and a generation module for Chemma-RC to enhance its compatibility with different chemical reaction conditions. Prediction modules are used to generate probability distributions over potential tokens, and we define two types of loss for this:

$$\text{Prediction}: \begin{cases} (1) \; X, G, T \xrightarrow{(classifier)} (c_i, \widehat{c}_i) : \mathcal{L} = \sum_{i \in I} CrossEntropyLoss\,(c_i, \hat{c}_i) \\ (2) \; X, G, T \xrightarrow{(generate)} (C, \widehat{C}) : \mathcal{L} = -\sum_{l=1}^{L} \sum_{v=1}^{V} y_l^v \log P_\theta(y_l^v \mid y_{<l}, (x, g, t)) \end{cases} \tag{3}$$

where $classifier$ refers to classification head, $I$ is the chemical context condition number, $c_i$ is the predicted label of the $i$-th condition, $\widehat{c}_i$ is the ground truth label of the $i$-th condition; $generate$

refers to generation head, $C$ and $\widehat{C}$ are the combination of predicted and the ground truth conditions, respectively. $L$ is the sequence length, $V$ is the vocabulary size. $y_l$ is the one-hot encoded target token at position $l$, $y_l^v$ is the $v$-th element of the one-hot encoded target token at position $l$; $y_{<l}$ represents all previous tokens before position $l$; $(x, g, t)$ is the input context tokens representing SMILES, graphs, and corpus.

### 3.2.3 MODALITY PROJECTION

For the reaction condition recommendation task, the representation of the reaction is extracted by encoders (see in section 3.2.2), and LLMs tokenize the text representation. However, fusing two types of representation introduces inductive biases issues (Baltrušaitis et al., 2018; Jaegle et al., 2021). To effectively fuse representations from multiple modalities, we propose the Perceiver (Jaegle et al., 2021) module for modality projection, seen 'modality projection' in Figure 1. This module employs latent tokens to align graphs and SMILES embeddings with text-related tokens extracted from question prompts and a text-augmented corpus. During training, we employ two transformer-based Perceivers as projectors. Although these modules share an identical model architecture, they are distinguished by their unique weights. Consequently, learnable tokens contain highlighted reaction cues that are most related to the text tokens. We show the pseudo-code for modality projection in Appendix. C.

## 4 EXPERIMENTS AND RESULTS

### 4.1 DATA

We curate two large datasets, named USPTO-Condition and USPTO_500MT_Condition for evaluation. Data volumes are presented in Table. 1. The visualization of data distribution is depicted in Figure. 4. As depicted in Table. 8, for the USPTO-Condition dataset, five conditions categories are separated by commas in order. For the USPTO_500MT_Condition dataset, all conditions are combined by dot as strings. The detailed data description can be seen in Appendix. B.

Table 1: Data description of USPTO-Condition and USPTO_500MT_Condition.

| Dataset | Sample of conditions | Prediction type | Training set |
|---|---|---|---|
| USPTO-Condition | [Zn],C1CCOC1,O,CO,[Cl-].[NH4+] | classification | 546,728 |
| USPTO_500MT_Condition | CO.[Na+].CC(=O)O.[BH3-]C#N | generation | 88,410 |

### 4.2 EXPERIMENT SETUP

In our work, the reaction encoder is implemented based on Wang et al. (Wang et al., 2023a). A pretrained graph model proposed by (Schlichtkrull et al., 2018) encodes the molecules in the reaction. We utilize LLaMA-2 (Touvron et al., 2023) as a text decoder. Each reaction has the corresponding corpus, a paragraph describing a chemical reaction with an average length of 190 tokens. During the training process, we fix the weight parameters of GCN, reaction encoder, and LLaMA-2. The modality projection and condition prediction layer is trainable. The trainable parameters constitute approximately 0.3 billion out of the total 7 billion parameters. The training process is conducted with a batch size of 16 for fewer than 6 epochs over 48 hours, utilizing a GPU configuration of 2×48 GB NVIDIA A6000 GPUs. The inference process is highly efficient and can be performed using a single 80 GB NVIDIA A800 GPU. The detailed training setting can be seen in Appendix. A.

### 4.3 PERFORMANCE COMPARISON

We assess the performance of our proposed Chemma-RC for reaction condition recommendation. The top-$N$ accuracy of condition recommendation on the combined test datasets of USPTO-Condition and USPTO_500MT_Condition are presented in Table. 2 and Table. 3, respectively. Compared methods include rxnfp LSTM (Gao et al., 2018), Reaction GCNN (Maser et al., 2021), TextReact (Qian et al., 2023), and Reagent Transformer (Andronov et al., 2023). The details of the baselines are present in Appendix. D.

Table 2: Results of reaction condition recommendation on USPTO-Condition dataset. The best performance is in **bold**.

| Model | Top-$k$ Accuracy (%) | | | | | | | | | | | | | | |
| | Catalyst | | | Solvent 1 | | | Solvent 2 | | | Reagent 1 | | | Reagent 2 | | |
| | 1 | 3 | 5 | 1 | 3 | 5 | 1 | 3 | 5 | 1 | 3 | 5 | 1 | 3 | 5 |
|---|---|---|---|---|---|---|---|---|---|---|---|---|---|---|---|
| rxnfp LSTM | 91.6 | 94.1 | 95.2 | 48.3 | 64.4 | 70.2 | 81.4 | 83.4 | 84.6 | 48.2 | 64.4 | 70.8 | 76.5 | 84.1 | 86.4 |
| Parrot | 89.9 | 96.4 | 97.7 | 35.2 | 60.9 | 72.2 | 81.2 | 93.7 | 96.7 | 40.4 | 62.3 | 71.7 | **80.6** | 90.6 | 93.6 |
| TextReact$_s$ | 92.1 | 98.0 | 99.1 | 51.4 | 68.5 | 79.3 | 81.6 | 93.4 | 96.9 | 51.1 | 69.6 | 79.1 | 77.9 | 91.1 | 94.9 |
| Chemma-RC$_s$ | **92.8** | **98.6** | **99.3** | **54.7** | **76.5** | **84.9** | **81.9** | **94.8** | **97.6** | **53.4** | **75.9** | **83.9** | 78.6 | **93.2** | **96.2** |

For the USPTO-Condition dataset, we calculate top-$k$ accuracy with a strict matching policy. As depicted in Table. 2, TextReact$_s$ refers that we utilize *similar text* (Qian et al., 2023) paired with the corresponding reaction for training. To avoid label leak issues, we do not use *gold text* mentioned in his work for training or testing. Chemma-RC$_s$ refers that we use a similar corpus paired with each reaction as input to construct Q&A instruction datasets for training. Thanks to the work of Qian et al., we can retrieve the most similar corpus for each reaction from the literature or patents using their pre-trained model.

Table 3: Results of reaction condition recommendation on USPTO_500MT_Condition dataset. The best performance is in **bold**.

| Model | Top-$k$ Accuracy (%) | | | |
| | 1 | 3 | 5 | 10 |
|---|---|---|---|---|
| Reagent Transformer | 17.5 | 27.5 | 31.6 | 35.6 |
| Reaction GCNN | 16.1 | 27.5 | 33.0 | 40.2 |
| Parrot | 13.8 | 25.3 | 31.4 | 37.9 |
| nach0 | 13.1 | - | - | - |
| **Chemma-RC** | **25.9** | **47.2** | **67.8** | **79.2** |

From the results, we observe that due to the low data sparsity of catalysts in the USPTO-Condition dataset (Figure. 9), all compared methods perform well, with the top-1 accuracy of the catalyst almost exceeding 90%. For solvent prediction, Chemma-RC outperforms the other methods, with top-1 accuracy of 54.7% (solvent 1) and 81.9% (solvent 2), respectively. The overall top-1 accuracy of Chemma-RC is 34.1% higher than that of the Parrot model. It can be concluded that our proposed Chemma-RC exhibits strong capabilities of reaction representation, akin to the learning process of chemists (Achiam et al., 2023).

Unlike the USPTO-Condition dataset which includes three types of chemical condition data–catalysts, solvents, and reagents–the USPTO_500MT_Condition dataset categorizes all conditions as 'reagents'. The performance of comparative methods on the USPTO_500MT_Condition dataset is shown in Table. 3. We have broadened several sets of baseline models to illustrate the feasibility of Chemma-RC, including nach0 (Livne et al., 2024), transformer-based models (Andronov et al., 2023), and other methods. The visualization of performance is shown in Appendix Figure. 6. We examine top-1, top-3, top-5, and top-10 predictive results. Notably, for USPTO_500MT_Condition datasets (Table. 3), we can see that Chemma-RC demonstrates the most favorable performance, where achieves 25.9% top-1 accuracy when compared with other baseline methods such as Reagent Transformer (17.5%), Reaction GCNN (16.1%), nach0 (13.1%). All SMILES conditions in the USPTO_500MT_Condition dataset are concatenated with dots, resulting in challenges due to the lengthy token sequences. However, Chemma-RC, pre-trained on a vast natural language corpus, effectively manages and accurately generates these long tokens.

## 4.4 ABLATION STUDY

### 4.4.1 MODEL STRUCTURE

In Chemma-RC, SMILES strings provide a textual representation of molecular structures, concisely encoding vital connectivity and stereochemistry details. Structural graphs of molecules offer a topological view of molecules in two-dimensional space, where atoms are nodes and bonds are edges. The textual corpus introduces a natural language context into the model to enhance the chemical interpretation capability of LLMs.

First, to examine the effect of different modalities on the performance of Chemma-RC, we evaluate the performance under the different combinations of mono-domain data including SMILES, graph, and corpus on the USPTO-Condition dataset. As indicated in Table. 4, from the results, we can see that different mono-domain data have different contributions for the entire performance. For the

Table 4: Performance evaluation of Chemma-RC under different combinations of mono-domain data on the USPTO-Condition Dataset.

| SMILES | Graph | Corpus | Top-$k$ Accuracy (%) | | | | | | | | | | | | | | |
|---|---|---|---|---|---|---|---|---|---|---|---|---|---|---|---|---|---|
| | | | Catalyst | | | Solvent 1 | | | Solvent 2 | | | Reagent 1 | | | Reagent 2 | | |
| | | | 1 | 3 | 5 | 1 | 3 | 5 | 1 | 3 | 5 | 1 | 3 | 5 | 1 | 3 | 5 |
| ✓ | ✗ | ✗ | 90.3 | 97.5 | 98.7 | 37.1 | 64.5 | 75.7 | 80.8 | 92.9 | 96.8 | 37.1 | 63.5 | 74.7 | 73.7 | 89.9 | 94.1 |
| ✗ | ✓ | ✗ | 87.1 | 93.3 | 95.5 | 15.3 | 40.5 | 58.2 | 80.7 | 91.9 | 95.5 | 34.6 | 56.8 | 67.5 | 75.4 | 86.6 | 90.6 |
| ✗ | ✗ | ✓ | 87.1 | 87.4 | 87.8 | 14.1 | 26.1 | 44.9 | 80.7 | 88.1 | 92 | 26.0 | 32.1 | 37.3 | 75.1 | 76.6 | 77.9 |
| ✓ | ✗ | ✓ | 92.6 | 98.5 | **99.3** | 54.0 | 76.0 | 84.4 | **81.8** | 94.7 | **97.6** | 52.8 | 75.4 | 83.3 | 78.6 | 93.1 | 96.1 |
| ✓ | ✓ | ✗ | 91.3 | 98.1 | 99.1 | 42.1 | 68.8 | 79.4 | 80.1 | 93.5 | 97.1 | 45.2 | 70.4 | 79.9 | 76.7 | 91.4 | 95.1 |
| ✓ | ✓ | ✓ | **92.7** | **98.6** | 99.2 | **54.6** | **76.4** | **84.8** | **81.8** | **94.8** | **97.6** | **53.4** | **75.8** | **83.9** | **78.7** | **93.2** | **96.2** |

prediction of solvent 1, which is the most challenging task, the model enhanced with SMILES representation (first row) outperforms the models trained solely on graph-based features (second row) and corpus data (third row), achieving 21.8% and 23.0% higher top-1 accuracy, respectively. Subsequently, we investigate how chemical mono-domain data combination affects model performance compared to individual types of data (fourth row to sixth row). **By incorporating a corpus into the model already trained with SMILES representations, we achieve a 16.9% improvement in solvent 1 top-1 prediction accuracy.** Similarly, **integrating graph features into the SMILES-based model results in a 5.0% improvement in solvent 1 top-1 accuracy.** The effectiveness of incorporating additional corpus data and SMILES representations can be attributed to the LLM's pre-training on extensive SMILES sequences and reaction data, which equips it with a more comprehensive understanding of chemical reactions and enhances its performance on RCR tasks. In a word, experimental results substantiate that integrating different modalities of chemical data including SMILES, graphs, and natural corpus, presents an effective representation of reactions, which is effective for RCR scenarios.

### 4.4.2 DATA SPLIT STRATEGY

We include the other baseline methods for comparison on the USPTO-Condition dataset. We also evaluate Chemma-RC's performance under different dataset splitting strategies, including random split (RS) and time-based split (TS), to further demonstrate its robustness across diverse conditions. A detailed introduction of each method and experiment settings are illustrated in the Appendix. D. TextReact (gr) refers to the TextReact model without retrieving gold texts for testing. From the results, we can see that the performance of other baseline models such as rxnfp LSTM (Gao et al., 2018), rxnfp retrieval, Transformer, and ChemBERTa (Chithrananda et al., 2020) shows moderate success. However, these models consistently deliver lower accuracy rates compared to TextReact (gr) and Chemma-RC. Chemma-RC significantly outperforms all baseline methods across both RS and TS settings. Notably, it achieves a Top-1 (RS) accuracy of 72.3%, which is substantially higher than the second-best approach, TextReact (gr), at 47.2%.

Table 5: Evaluation results for reaction condition recommendation (RCR). RS: random split; TS: time split. Scores are accuracy in %.

| | RCR (RS) | | | | RCR (TS) | | | |
|---|---|---|---|---|---|---|---|---|
| | Top-1 | Top-3 | Top-10 | Top-15 | Top-1 | Top-3 | Top-10 | Top-15 |
| rxnfp LSTM | 20.5 | 30.7 | 41.7 | 45.3 | 15.2 | 26.2 | 40.7 | 45.4 |
| rxnfp retrieval | 27.2 | 37.5 | 47.9 | 51.1 | 7.8 | 15.2 | 27.3 | 31.5 |
| Transformer | 30.0 | 43.8 | 56.7 | 60.5 | 18.7 | 31.8 | 47.6 | 52.7 |
| ChemBERTa | 30.3 | 44.7 | 58.0 | 62.0 | 18.7 | 31.9 | 47.6 | 52.8 |
| TextReact(gr) | 47.2 | 59.9 | 65.0 | 71.4 | 36.3 | 50.4 | 56.2 | 63.8 |
| Chemma-RC | **72.3** | **87.8** | **92.4** | **96.5** | **69.6** | **86.7** | **91.7** | **96.2** |

### 4.4.3 MODALITY PROJECTION

By leveraging the strengths of multiple modalities, multimodal LLMs can achieve higher accuracy in a wide range of applications. However, aligning representations among different modalities remains a challenging task. In our proposed Chemma-RC, we employ the Perceiver module (Jaegle et al.,

2021) to integrate molecular SMILES tokens and graphs tokens into text-related language space, where text tokens are augmented by the reaction corpus, as illustrated in Figure 1. This modality projection module maps the embeddings of reactions to a latent vector and enhances this representation using a Transformer tower. Consequently, learnable queries contain highlighted reaction contents that are most related to the text tokens. We compared three typical methods for modality projection, including Perceiver (Jaegle et al., 2021), Reprogramming (Jin et al., 2024), and MLP.

Table 6: Performance evaluation of Chemma-RC under different modality projections, the best performance are in bold.

| Projection Layer | Top-$k$ Accuracy (%) | | | | | | | | | | | | | | |
| | Catalyst | | | Solvent 1 | | | Solvent 2 | | | Reagent 1 | | | Reagent 2 | | |
| | 1 | 3 | 5 | 1 | 3 | 5 | 1 | 3 | 5 | 1 | 3 | 5 | 1 | 3 | 5 |
| MLP | 90.9 | 97.8 | 98.9 | 51.1 | 73.3 | 82.2 | 81.1 | 93.9 | 97.1 | 47.4 | 71.0 | 79.9 | 77.0 | 91.7 | 95.2 |
| Reprogramming | 92.1 | 98.3 | 99.1 | 52.8 | 75.1 | 83.7 | 81.3 | 94.3 | 97.4 | 50.2 | 73.5 | 81.9 | 77.7 | 92.5 | 95.7 |
| Perceiver | **92.7** | **98.6** | **99.2** | **54.6** | **76.4** | **84.8** | **81.8** | **94.8** | **97.6** | **53.4** | **75.8** | **83.9** | **78.7** | **93.2** | **96.2** |

As depicted in Table. 6, the Perceiver module achieves significant gains in the prediction of all categories. Compared with Chemma-RC (with Reprogramming), Chemma-RC (with Perceiver) can be further enhanced and attains peak performance in all predicted categories with 7.1% significant gain. Specifically, For the solvent 1 prediction, a hard case, the Perceiver module stands out with a top-1 accuracy of 54.6%, significantly surpassing MLP (51.1%) and Reprogramming (52.8%). Its ability to consistently achieve high accuracy in both top-1 and top-k evaluations suggests a robust and versatile approach for reaction condition recommendation.

## 4.5 TRANSFERABILITY EVALUATION ON HIGH-THROUGHPUT EXPERIMENTATION REACTION

Discovering effective reaction conditions precisely for high-throughput reaction condition screening is very important, as it has the potential to release chemists from laborious and costly trial-and-error workflows. Thus, we illustrate the transferability of our models through zero-shot evaluation on distinct high-throughput experimentation (HTE) datasets. We expect that Chemma-RC recommends conditions that yield high-product outputs. We select the Imidazole C–H arylation dataset extracted from the work proposed by Shields et al. in 2021 (Shields et al., 2021) for evaluation, where the substrate scope contains 8 imidazoles and 8 aryl bromides associated with conditions including ligands, bases, and solvents.

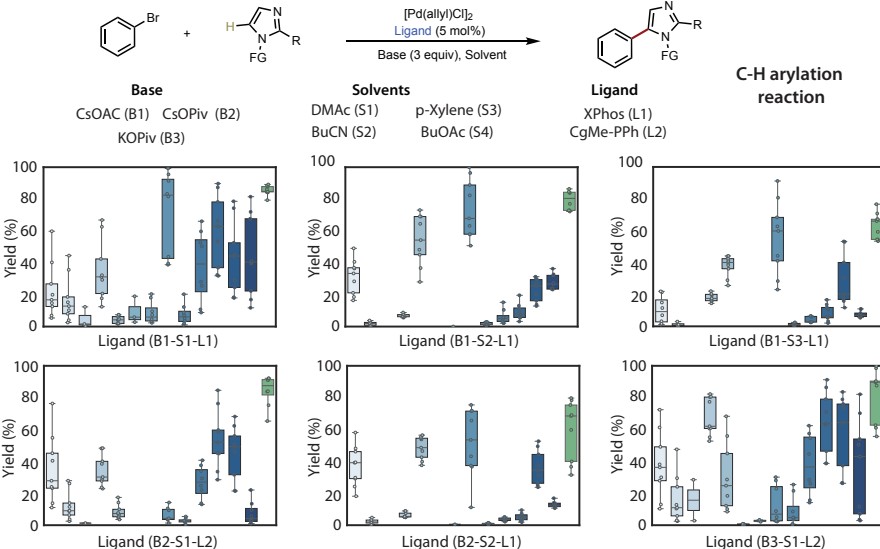

Figure 3: Boxplot of the performance for ligand recommendation on C-H arylation reaction.

Catalysts are vital compounds in chemical reactions, as they play a crucial role in determining both reactivity and yield. The catalyst used in imidazole C–H arylation comprises a metal (Pd) and ligands. Thus, we evaluate the performance of ligand recommendations. First, we ensure that reaction

data of imidazole C–H functionalization is excluded from the test set of the USPTO-Condition dataset to prevent data leakage issues. Chemma-RC recommends a ligand under a pre-defined solvent-base combination of conditions. As shown in Figure. 3, we randomly select six cases for performance evaluation. The referenced bases, solvents, and ligands can be found in the reaction formula, which has been annotated by 'B','S', 'L'. For example, in Figure. 3, under the combination of CsOAc and DMAc, Chemma-RC identifies the XPhos ligand, which results in a higher yield.

For recommended results (Figure. 10, Figure. 11) we can observe that, for **15** of the 16 base-solvent combinations, the recommended ligand performs best in terms of the median value of reaction yields, suggesting that Chemma-RC can recommend ligands with higher yields.

Moreover, we can conclude that the capability of Chemma-RC to recommend suitable conditions for chemical reactions has the potential to accelerate high-throughput reaction condition screening in the future.

## 5    Conclusion and Limitations

**Conclusions** In this paper, we present a multimodal LLM, a.k.a. Chemma-RC for chemical reaction condition recommendation. Trained with 1.2 million pair-wised Q&A instruction datasets that integrate with multimodal reaction representations and corpus in natural language, Chemma-RC effectively answers questions regarding reaction conditions through either a classification head or sequence generation.

**Limitations** Further, we will focus on how the token length of each modality improves its performance across various chemical reaction tasks in future work.

## 6    Reproducibility Statement

To ensure the reproducibility of our work, we have used datasets which have been published in (Wang et al., 2023a; Lu & Zhang, 2022), and the data links are as follows: USPTO_500MT_Condition and USPTO-Condition. Additionally, we commit to releasing the full implementation of our code, including model architectures, training pipelines, and evaluation scripts, upon acceptance and publication of this paper. Detailed instructions and necessary dependencies are provided in the Appendix to facilitate easy reproduction of our results.

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

## APPENDIX

## A  TRAINING SETTINGS

To realize peak efficiency within our Chemma-RC model, we carefully design the training phases. This section offers a comprehensive summary of the training settings and the hyperparameter values. Through the detailed orchestration of these parameters, we ensure that Chemma-RC is capable of fully leveraging its capabilities in the application contexts.

- **Optional Settings:** There are alternatives for modification in the Chemma-RC framework, such as the replacement of the Perceiver-based modality projection layer with other architectures like Reprogramming and MLP.

- **Reaction Condition Recommendation task:** Within the framework, the model takes the 32-layer LLaMA-2-7b as the LLM backbone. Besides, we utilize a pre-trained SMILES-to-text retriever proposed by Qian et al. (Qian et al., 2023) and extract the most similar unpaired corpus as the reaction text. Meanwhile, we introduce Parrot, a Bert-like model to encode the reaction SMILES. We leverage R-GCN (Schlichtkrull et al., 2018) to encode the molecules in the reaction, and the combination of reactant and product embeddings is considered as the reaction representation. In the training process, the encoders in all modalities are frozen. After the alignment of the representation space, the SMILES and the graph-based tokens have a length of 128 and 3, respectively. Additionally, the model

employs the OneCycleLR as the learning rate schedular, initializing the learning rate as 3e-5. The batch size is set to 16, with less than 6 epochs 48 hours in training. The GPU configuration is $8 \times 80G$ A800.

# B  DATA DESCRIPTION

We curate two large datasets, named USPTO-Condition and USPTO_500MT_Condition, with the data volumes presented in Table. 8. Both datasets are split with the ratio of train:validation:test as 8:1:1 in our work. For USPTO-Condition dataset, all molecules including reactants, products, and conditions are collected in canonical SMILES. Each reaction entry contains five condition labels, including one catalyst, two solvents, two reagents, and an additional "none" category is introduced to illustrate that the reaction does not require this type of reaction condition (Gao et al., 2018). The visualization of data distribution is depicted in Figure. 4 (left). From Figure. 4 we can see that this dataset covers a vast variety of reaction types, characterized by a substantial proportion of heteroatom alkylation, arylation, and acylation reactions, while C-C formation reactions are less included. We also introduce the corpus of reaction descriptions proposed by Qian et al. (Qian et al., 2023) into the USPTO-Condition dataset. Each reaction is associated with a corpus of reaction descriptions. It should be noted that the corpus will not be utilized directly for training. Instead, we employ the corpus as an input for the pre-trained retrieval module proposed by (Qian et al., 2023). This approach allows us to obtain similar embeddings necessary for the multimodal representation learning of our Chemma-RC, and avoid data leaking issues. For USPTO_500MT_Condition datasets, it collects top-500 types of reactions from the USPTO-MIT datasets (Coley et al., 2017), in which the top-100 types of reactions make up 59% of the entire dataset, which can be seen in Figure. 4 (right). In order to calculate the predicted accuracy on the USPTO_500MT_Condition dataset, it is necessary to separate all reagents in an appropriate manner. However, separating reagents using the dot as a delimiter is challenging, as compounds like [Na+].[OH-] constitutes a single reagent and cannot be split. Besides, to have a comprehensive knowledge of the datasets, we do sparsity analyses. We calculate the non-empty count and density of every condition in the USPTO-Condition dataset, which is presented in Table. 9. From the table, we can see that some conditions, such as 'Catalyst', 'Solvent 2', and 'Reagent 2' show a high extent of sparsity, with a non-empty density of fewer than 30%. For the USPTO_500MT_Condition, as it only covers the condition of non-split reagents, all of the reaction entries have their corresponding non-empty condition label.

Furthermore, we make an investigation on the condition categories in the USPTO-Condition and USPTO_500MT_Condition dataset, which is illustrated in Figure. 5. The visualization of the most common chemical contexts of the regents, catalysts, and solvents in USPTO-Condition, and separate reagents in USPTO_500MT_Condition is depicted in Figure. 5 (A-D), respectively. From the figures, we learn that reaction conditions have a property of diversity and imbalance. Besides, we count categories of every condition, as is presented in Figure. 5 (E). Reagents in both datasets consist of more than 200 categories, which highlights the difficulty of the reaction condition recommendation task. Additionally, we prove that reagents in the USPTO_500MT_Condition dataset follow the power-law distribution, which indicates the condition keeps the long-tail feature in distribution and a small number of categories account for the majority of the data size.

Table 7: Question templates generated by GPT-4.

| Task | Description |
|---|---|
| Solvent prediction | Could you suggest potential solvents that could have been used in the given chemical reaction, taking into consideration their polarity and compatibility with the reactants? |
| Reagent prediction | Please suggest some possible reagents that could have been used in the following chemical reaction. |
| Catalyst prediction | Considering the chemical reaction in question, which catalysts could be effective? |
| Condition prediction (all) | Given the current chemical reaction, what would be the appropriate conditions to consider? |

Table 8: Data volume of USPTO-Condition and USPTO_500MT_Condition datasets.

| Dataset | Training set | Validation set | Testing set |
|---|---|---|---|
| USPTO-Condition | 546,728 | 68,341 | 68,341 |
| USPTO_500MT_Condition | 88,410 | 9,778 | 10,828 |

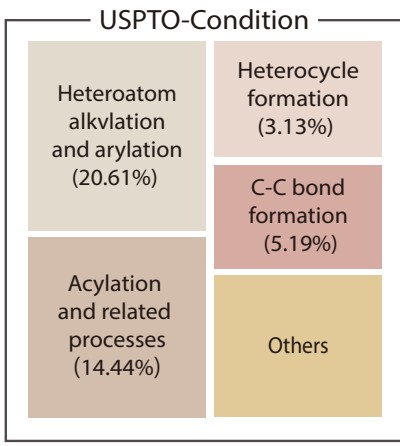 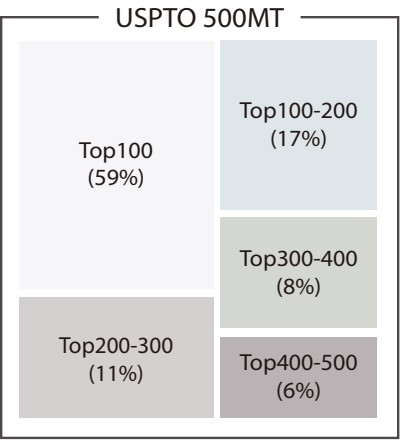

Figure 4: Left: The reaction distribution of USPTO-Condition. Right: The reaction distribution of USPTO_500MT_Condition.

## C  DETAILS OF MODALITY ALIGNMENT

For the reaction condition recommendation task, the representation of the reaction is extracted by encoders (see in section 3.2.2), and the text representation is tokenized by LLMs. However, fusing two types of representation introduces inductive biases issues (Baltrušaitis et al., 2018; Jaegle et al., 2021). To effectively fuse representations from multiple modalities, we propose the use of a projection module, the Perceiver (Jaegle et al., 2021), for modality alignment (Figure 1). This module employs latent queries to align graph and SMILES tokens with text-related tokens, such as question prompts and a text-augmented corpus. We show the pseudo-code for modality projection in Algorithm. 1.

## D  MODEL PERFORMANCE

A chemical reaction can be represented as the transformation of a sequence of characters (reactants, conditions) into another sequence (products), with compounds connected by special characters, such as '>>'. This structure makes sequence-to-sequence models, such as the Transformer, well-suited for predictive modeling of reaction representation (Schwaller et al., 2019; Irwin et al., 2022). However, existing SMILES-based Transformer models for reaction representation encounter limitations in various aspects, particularly with respect to atom permutations and the interpretability of reaction mechanisms. Consequently, our proposed Chemma-RC fuses data from diverse sources including corpus, SMILES and graphs of molecules to present a comprehensive view of the reaction. We assess the performance of our proposed Chemma-RC and the aforementioned baseline methods for reaction condition recommendation. The top-$N$ accuracy of condition recommendation on the combined test datasets of USPTO-Condition and USPTO_500MT_Condition are presented in Table. 2

Table 9: Sparsity analysis of the USPTO-Condition dataset.

| **USPTO-Condition** | Catalyst | Solvent 1 | Solvent 2 | Reagent 1 | Reagent 2 |
|---|---|---|---|---|---|
| **Non-empty count** | 89,756 | 673,634 | 130,326 | 504,169 | 170,752 |
| **Non-empty density** | 13% | 99% | 19% | 74% | 25% |

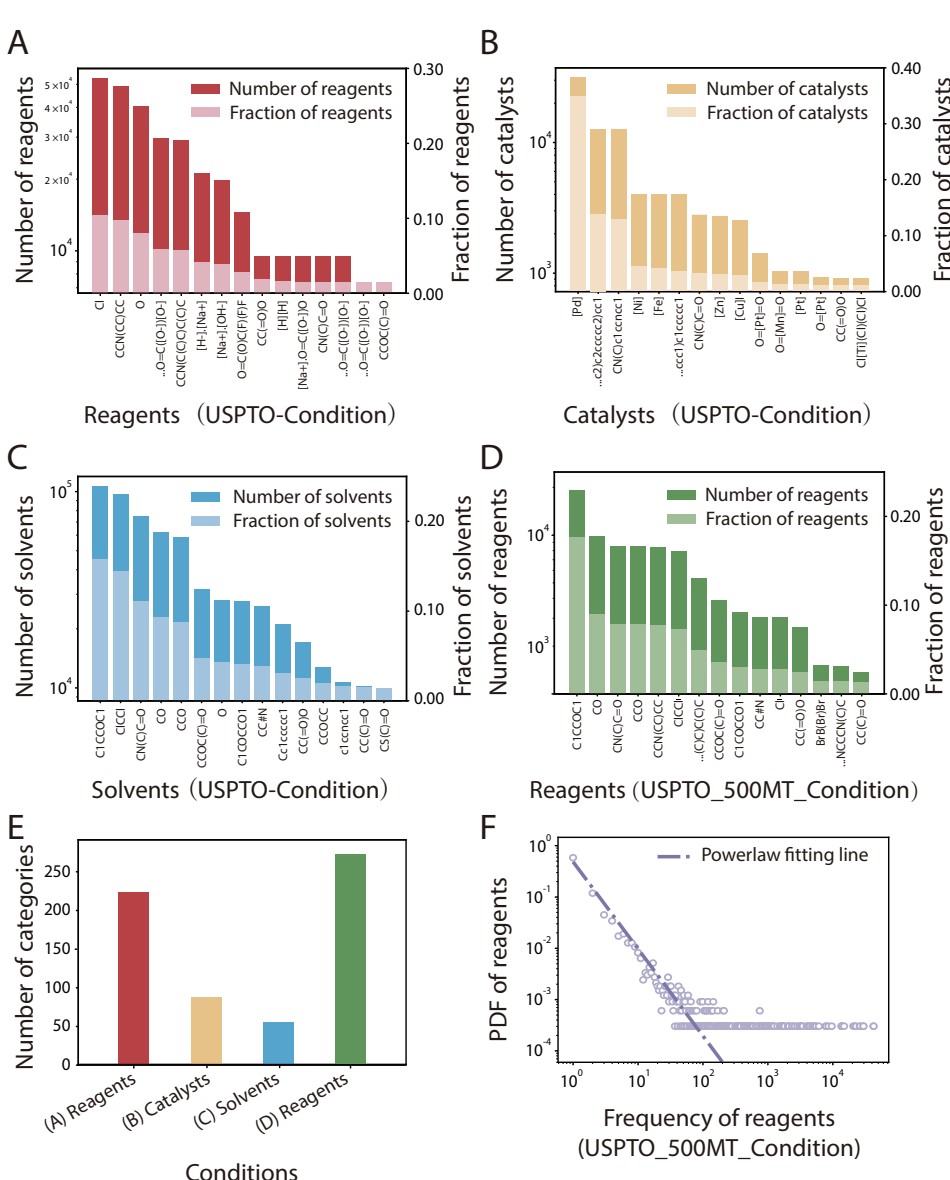

Figure 5: **Distribution of types of reactions in the USPTO-Condition and USPTO_500MT_Condition.** **(A-D)** The bar charts of the fifteen most common reagents, catalysts, and solvents in the USPTO-Condition and reagents in the USPTO_500MT_Condition, respectively, where the shallow color presents the decimal-scale proportion and the deep color presents the log-scale count. **(E)** The bar charts of the total category count of the conditions illustrated in (A-D). **(F)** Power law fitting of the reagent distribution in the USPTO_500MT_Condition, where the shallow points show the probability density and the deep dashed-line shows the ideal power-law fitting, respectively.

---

**Algorithm 1** Pseudo code for modality projection.

---

`word_proj`, `perceiver_proj`: predefined linear and transformer-based projectors, respectively.

```
# B: batch size; C: channel size; n: content shape
# M: query length; N: shape of flatten reaction tokens;
# text_q: text query in shape (B, M, C)
# react_embed: reaction embedding in shape (B, N, C)
# word_embed: word embedding in shape (B, vocab_size, C)

# Key part 1: map transformer-based reaction feature
word_embed = self.word_proj(word_embed)
word_embed = word_embed.repeat(react_embed.size()[0], 1, 1)
react_embed = torch.cat([react_embed, word_embed], dim=1)
smiles_react_tokens = linear_layer(react_embed) # to make 128
    tokens

# Key part 2: map graph-based reaction features
graph_embed = self.word_proj(graph_embed)
graph_react_tokens = linear_layer(graph_embed) # to make 3 tokens

# Key part 3:
reaction_tokens = torch.cat([smiles_react_tokens,
    graph_react_tokens], dim=1)

# Key part 4: modality projection
reaction_tokens_from_smiles = self.perceiver_proj_smiles(
    smiles_react_tokens)
reaction_tokens_from_graphs = self.perceiver_proj_graphs(
    graph_react_tokens)

# concat token
final_token = torch.cat([reaction_tokens_from_smiles,
    reaction_tokens_from_graphs, text_q], dim=1)
```

---

and Table. 3, respectively. We introduce several comparative methods to illustrate the performance of Chemma-RC.

1. rxnfp LSTM (Gao et al., 2018). This method proposes a reaction fingerprint to represent the difference between the product and reactant fingerprints.

2. rxnfp retrieval. It uses the conditions of the most similar reactions in the training set as the prediction. Similar reactions are determined based on the $L_2$ distance of reaction fingerprints.

3. Transformer. It uses the same architecture as the TextReact predictor. This baseline represents the state-of-the-art model that only takes chemistry input.

4. ChemBERTa Chithrananda et al. (2020). It is same as the Transformer baseline except that the encoder is pre-trained on external SMILES data.

5. Reaction GCNN (Maser et al., 2021). This method proposes a machine-learned ranking model to predict the set of conditions used in a reaction as a binary vector.

6. Parrot (Wang et al., 2023a). This method leverages the attention-based model architecture to encode the reaction and design a training methodology specifically to enhance the reaction center.

7. TextReact (Qian et al., 2023). It aims to enhance the molecular representation of the reaction by introducing relevant corpus retrieved from literature into sequence-to-sequence Transformers.

8. Reagent Transformer (Andronov et al., 2023). This method leverages Molecular Transformer, (Schwaller et al., 2019) a state-of-the-art model to tackle the task of reagent prediction.

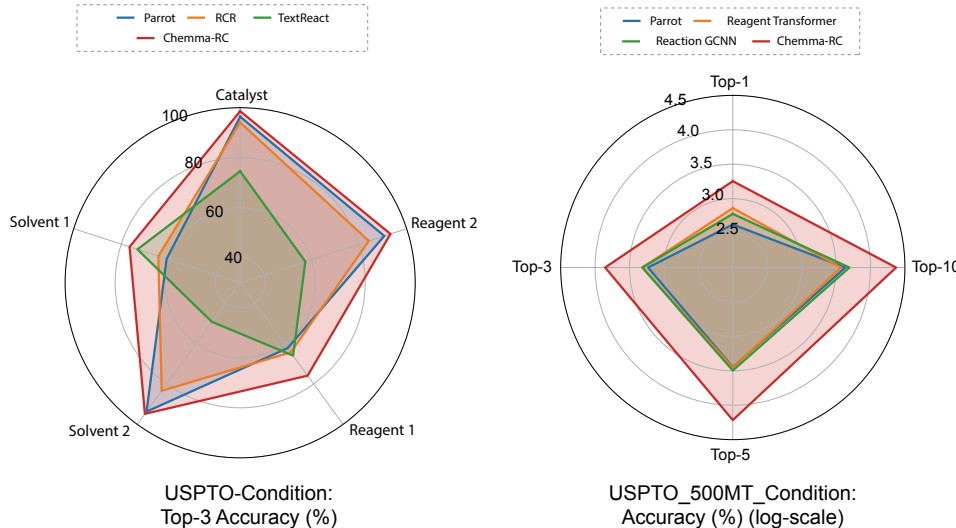

Figure 6: **Left:** Radar plot of top-3 predition accuracy of conditions on the USPTO-Condition dataset. The classification performance consists of comparative methods such as Parrot, RCR, TextReact, and our methods with similar corpus. **Right:** Radar chart of log-scale accuracy of reagents in the USPTO_500MT_Condition dataset.

To have a comprehensive overview of the recommendation performance, we visualize the prediction results of USPTO-Condition and USPTO_500MT_Condition datasets, as described in Table. 2, 3. Specifically, we draw radar charts of our model and other competitive models, which are presented in Figure. 6. For the USPTO-Condition dataset, we reproduce Parrot, RCR, and TextReact. Then, we plot the top-3 predicting accuracy of different conditions (catalyst, solvent 1, solvent 2, reagent 1, and reagent 2), as depicted in Figure. 6 (left). For the USPTO_500MT_Condition dataset, we recommend reagents in SMILES sequence and take Parrot, Reagent Transformer, and Reaction GCNN as comparative methods. For more intuition, we visualize top-1, 3, 5, and 10 exactly matched accuracy in log scale, which is shown in Figure. 6 (right). From the charts, we can see that our model covers the largest area of the performance circle in both datasets, indicating that Chemma-RC markedly outperforms other competitive models.

### D.1 GERALIZATION PERFORMANCE

In order to validate the out-of-domain performance of Chemma-RC, we employ Chemma-RC trained on the USPTO_500MT_Condition to test on the USPTO-Condition. The evaluation strategy includes three specific training conditions: reagents, catalysts, and solvents. We adopt a metric of **partial matched accuracy** to illustrate the generalization capability of Chemma-RC. Different from the complete matched accuracy that requires perfect matching between predictions and labels, the partial matched accuracy is more suitable to test the generalization capacity, which focuses more on whether the predicted results match a substitutable part of the ground truth. For example, if the predicted result is '[Na+].[OH-]' and the condition label is 'CO.[Na+].[OH-]', we consider that the prediction partially matches the ground truth, but not completely. The evaluation strategy includes three specific training conditions: reagents, catalysts, and solvents. Table. 10 reports the top-1 partial match accuracy for each condition prediction. From the results we can see that, Chemma-RC achieves a top-1 partial matched accuracy of 67.1% and 58.1%, respectively. This relatively high accuracy indicates that solvents and reagents have more consistent characteristics that the model can learn effectively from USPTO_500MT_Condition and apply to USPTO-Condition. In contrast, The model's performance in predicting catalysts demonstrates a lower top-1 partial match accuracy at 89.9%.

Chemma-RC can successfully distinguish reagents from the combination of all conditions in a reaction. Additionally, training Chemma-RC on USPTO-Condition, a larger chemical reaction dataset, further enhances its ability to akin chemical knowledge.

Table 10: The top-1 partial matched accuracy of Chemma-RC under OOD setting.

| Evaluation strategy (train → test) | Acc (%) |
| --- | --- |
| USPTO_500MT_Condition → USPTO-Condition (reagent) | 67.1 |
| USPTO_500MT_Condition → USPTO-Condition (catalyst) | 89.9 |
| USPTO_500MT_Condition → USPTO-Condition (solvent) | 58.1 |

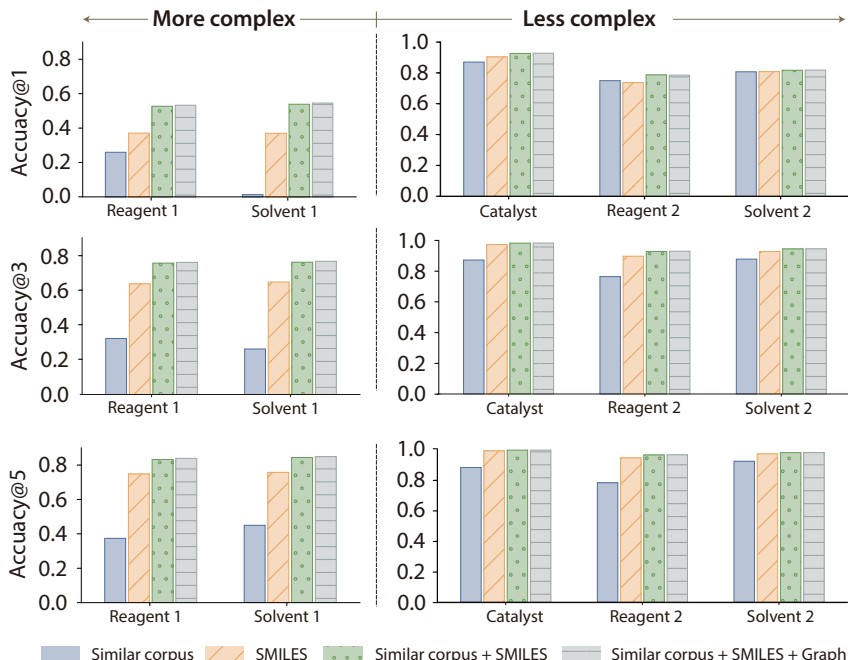

Figure 7: Bar charts demonstrating the ablation study of modalities including similar corpus, SMILES and graph. The classification performance is assessed on the conditions in the USPTO-Condition dataset, which are split into two groups according to data sparsity.

## D.2 ABLATION STUDY ON MODALITY

Besides, we visualize the results of the ablation study on modality on the USPTO-Condition dataset, which can be seen in Table. 4. Specifically, we categorize the conditions of the USPTO-Condition into two types: more complex and less complex. According to the data sparsity, reagent 1 and solvent 1 are considered more complex, while catalyst, reagent 2, and solvent 2 are considered less complex. Then, the investigation on the effectiveness of modalities comprising similar corpus, SMILES, graph is depicted in Figure. 7. From the results, we can see that compared with the model with multiple modalities, the model with single one modality degrades dramatically. Moreover, Chemma-RC with three modalities combined achieves the best performance, which demonstrates the vital importance of capturing the reaction representations from different dimensions.

## D.3 CASE STUDY

In this section, we select four cross-coupling reactions from USPO-Condition datasets for performance validation. We visualize the predicted results in Figure. 9. As depicted in Figure 9, the reaction centers and leaving groups are highlighted in different colors. For C–N cross-coupling reactions (the first and the third row), Chemma-RC can predict all conditions precisely. For C–C bond formation and Formylation reactions (the second and the fourth row), Chemma-RC fails to predict Ethyl Acetate (the second case) and THF (the fourth case). The reason why Chemma-RC is less effective for these reactions is that the data volume of C–C bond formation reactions in the USPTO-Condition dataset is only 5%, as shown in Figure 4. This limited representation constrains the

model's ability to learn the patterns associated with C–C bond formation reactions. Consequently, Chemma-RC lacks sufficient training examples to capture and generalize the underlying reaction mechanisms accurately. The scarcity of diverse and representative data hampers its effectiveness, leading to a lower precision in predicting these types of reactions.

Figure 8: Visualization of recommended conditions on four reactions. We select four Suzuki–Miyaura cross-coupling reactions to present the performance of condition recommendation. The reaction centers and leaving groups are highlighted in different colors.

Further, we visualize the predicted results on OOD datasets in Figure. 9. We select two reaction cases for analysis. In case 1, Toluene is not predicted by Chemma-RC. In case 2, 1,4-Dioxane and 1-(diphenylphosphaneyl)cyclopenta-2,4-dien-1-ide are predicted. However, it is confirmed that Toluene and 1,4-Dioxane are common solvents, and 1-(diphenylphosphaneyl)cyclopenta-2,4-dien-1-ide is frequently used as a ligand. Therefore, we do not categorize these as failed cases because the model successfully predicts all the reagents in the labels and avoids predicting other conditions.

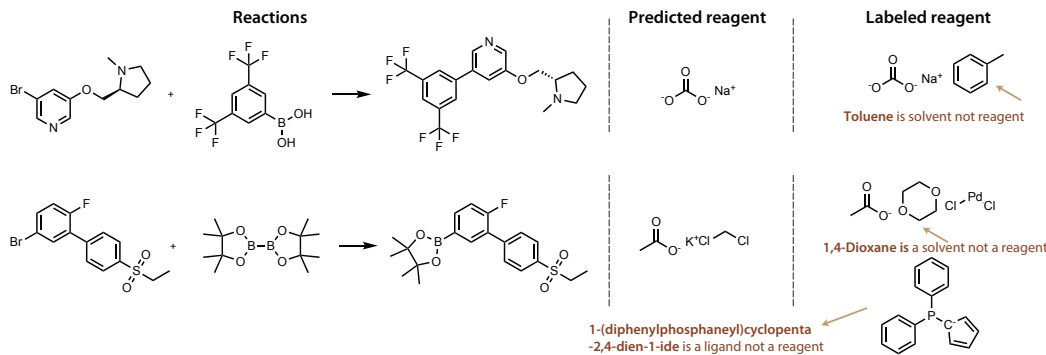

Figure 9: Visualization of recommended conditions on two reactions. In case 1, Toluene was not predicted by Chemma-RC. In case 2, 1,4-Dioxane and 1-(diphenylphosphaneyl)cyclopenta-2,4-dien-1-ide were predicted. However, it is confirmed that Toluene and 1,4-Dioxane are common solvents, and 1-(diphenylphosphaneyl)cyclopenta-2,4-dien-1-ide is frequently used as a ligand. Therefore, we do not categorize these as failed cases because the model successfully predicts all the reagents in the labels and avoids predicting other conditions.

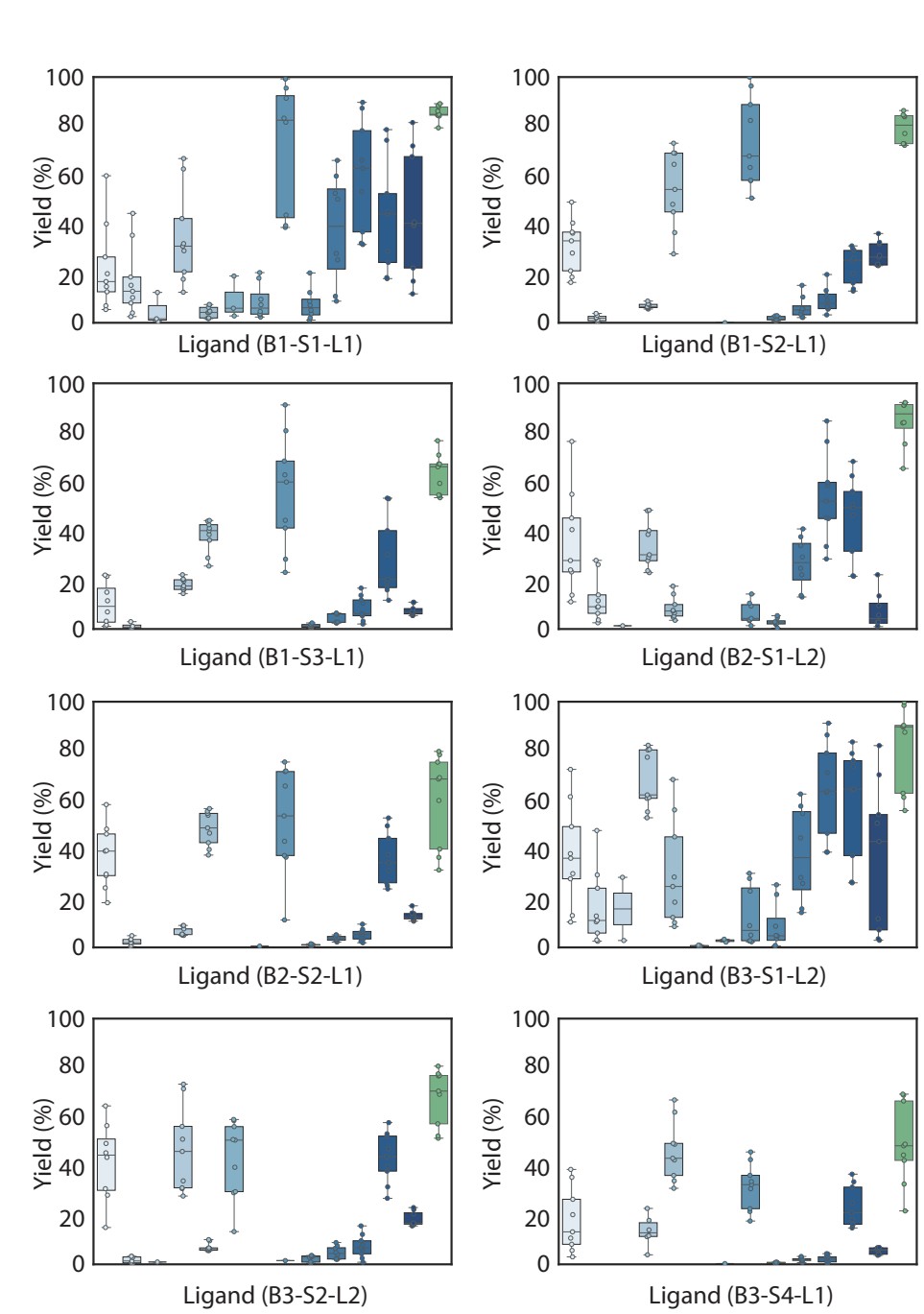

Figure 10: Boxplot of the performance for ligand recommendation (1).

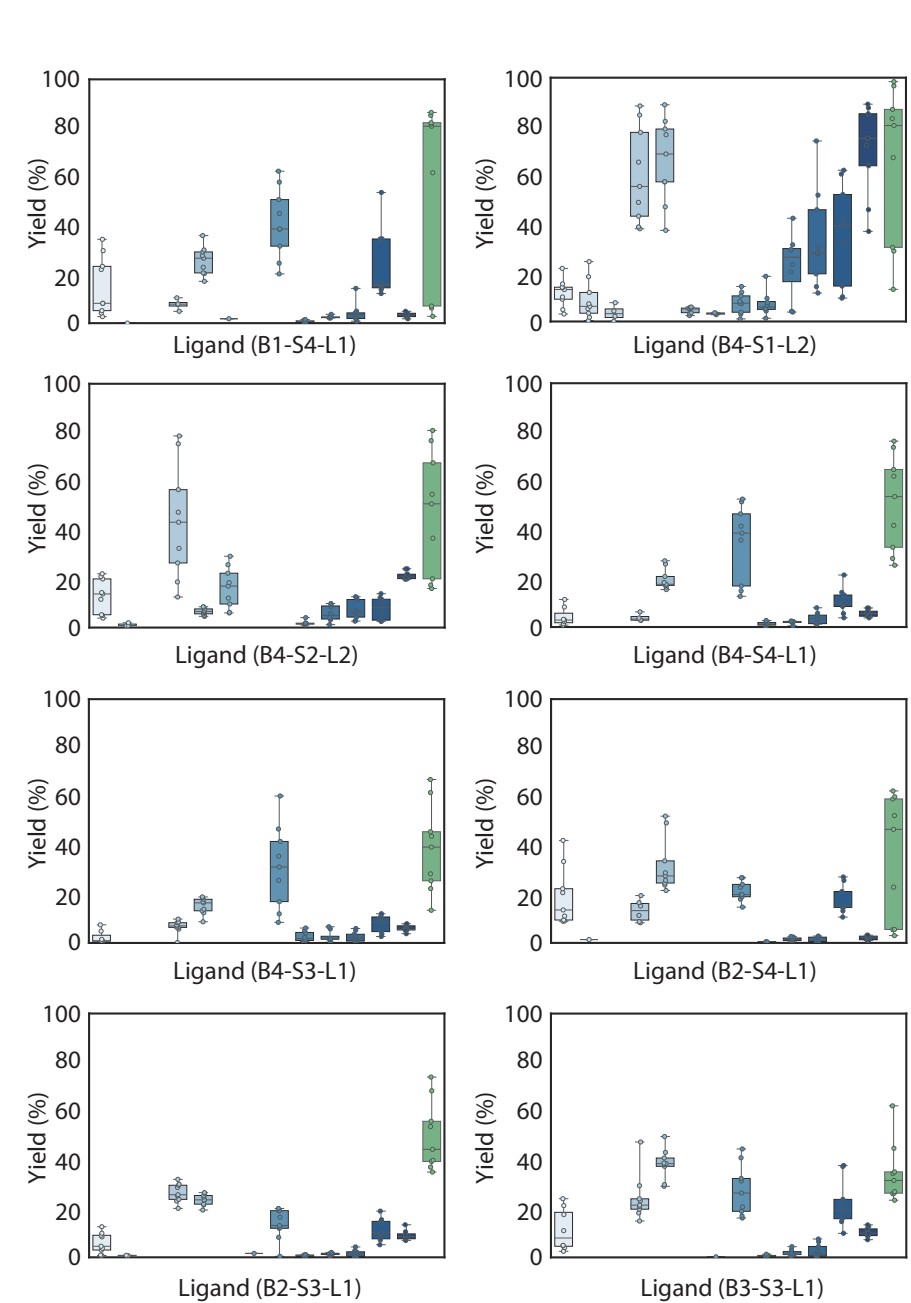

Figure 11: Boxplot of the performance for ligand recommendation (2).

