# OpenReview forum: "Text-Augmented Multimodal LLMs for Chemical Reaction Condition Recommendation"
_ICLR.cc/2025/Conference — ICLR 2025 Conference Withdrawn Submission_

### Official Review · Reviewer_od79 · 2024-10-28

**Soundness:** 2
**Presentation:** 2
**Contribution:** 2
**Rating:** 6
**Confidence:** 4

**Summary:**

The paper focuses on chemical reaction condition recommendation using LLMs. It proposes incorporating different data modalities, such as reaction formulas, graph-structured molecules, and reaction descriptions, into a single multimodal large language model (LLM) called Chemma-RC, fine-tuned through the Llama 2 model. The model is compared to previous baselines and shows promise in generating useful reaction conditions to accelerate high-throughput condition screening in chemical synthesis. Overall, the proposed method is interesting.

**Strengths:**

**Originality**: The paper focuses on an interesting problem with a reasonable solution. This is a useful and new case demonstrating the effectiveness of multimodal LLMs in drug discovery.

**Quality**: The proposed method seems promising.

**Clarity**: The paper includes figures to aid readability, with a well-illustrated method. The experimental setups and baselines are properly introduced.

**Significance**: The paper shows promising contributions to reaction condition recommendation.

**Weaknesses:**

1. Reaction condition recommendation is not a novel task in the era of LLMs. The paper needs a more thorough discussion and demonstration of why research focus should shift to LLM-based reaction condition recommendation. Do LLMs truly provide higher accuracy? Or do LLMs offer more interpretability through text generation? Has any work benchmarked the performance of LLMs in reaction condition recommendation?

2. Why are there two types of condition prediction modules mentioned in line 76? There is no motivation provided for them. LLMs are more naturally suited for generative tasks rather than prediction; why is prediction necessary here?

3. The ablation studies should be improved. In Table 4, consider adding a study that enables SMILES and graphs but disables the corpus (i.e., disabling the LLM part of the model).

4. There is a lack of detailed description regarding the process of constructing the question-answering dataset. Do the authors ensure that there is no data leakage? How was the data collected and deduplicated?

**Questions:**

1. In line 66: How can one determine that the LLMs are pre-trained on extensive reaction data?

2. Line 80: The term "Perceiver" appears without context. Please add a citation and provide an introduction.

3. In line 84/4.5: What distinguishes the USPTO 500MT Condition dataset from the USPTO-Condition dataset, and what causes their distributional differences? Are there any statistical results indicating that these tasks are out-of-distribution?

4. Figure 1 needs improvement. The boxes for data and model are indistinguishable, or there are no boxes, making it difficult for readers to determine where to begin.

5. Tables should be self-explanatory. In Table 1, what does the condition label mean?

6. In lines 300-303: Is there a specific reason why the authors freeze the parameters of all models and only fine-tune the projectors?

---

> ### Author Response · Authors · 2024-11-19
> **Response to Weakness 1-2 from Reviewer od79**
>
> #### Dear reviewer od79,
> #### We want to extend our heartfelt thanks for your supportive comments on our manuscript and the detailed suggestions.
> #### **Here is the response to Weakness 1:**
> #### Thank you so much for your opinions. As you said, several transformer-based models have already been developed for the reaction condition generation task, and achieve solid performance. In this study, we define large language models (LLMs) as autoregressive models pre-trained on massive natural language corpora. Based on this definition, we are the first to explore the application of LLMs to reaction condition recommendation (RCR). Previous work [1-2] has illustrated that to some extent, LLMs could suggest results in the realm of reagent selection on USPTO datasets, but they performed worse than traditional baseline models. Thus, here, we suppose that pre-trained with massive chemical reaction data including molecular simplified molecular-input line-entry systems (SMILES) in natural language, LLMs are endowed with fundamental chemical knowledge through text-to-text generation. These capabilities are especially valuable in chemistry. LLMs acquire knowledge through pre-training on natural language prediction tasks. Since reaction condition recommendation (RCR) can be formulated as a sequence-to-sequence generation problem, we hypothesize that training a language model on chemical reaction data can enable it to provide conditional recommendations for chemical reactions. Further, through our experiments, we confirm that the LLMs can be used to improve the performance of RCR, and our model performs notably better than current state-of-the-art baseline methods.
>
>
> |Model| Reactant selection | Solvent Selection | Ligand Section |
> |----- | ----- | ------ | ------ |
> |GPT-4 (zero-shot) | 0.299     | 0.526| 0.534 |
> |GPT-3.5 (zero-shot)| 0.400     | 0.368| 0.436 |
> |Davinci-003 (zero-shot) | 0.178     | 0.463 | 0.432 |
>
> #### [1] Guo T, Nan B, Liang Z, et al. What can large language models do in chemistry? a comprehensive benchmark on eight tasks[J]. Advances in Neural Information Processing Systems, 2023, 36: 59662-59688.
> #### [2] White A D. The future of chemistry is language[J]. Nature Reviews Chemistry, 2023, 7(7): 457-458.
>
> #### **Response to  weakness 2:**
> #### It's true that  LLMs are more naturally suited for generative tasks rather than prediction. However, we still have motivation to design two modules.
> #### - On the one hand, the motivation for performing classification tasks is to select the most suitable reaction conditions from commercially available libraries, as it is common practice to prioritize purchasable molecules.
> #### - On the other hand, the generation module can assist in designing novel molecules, which can be obtained by synthesis experiments conducted. Therefore, we define two distinct tasks including classification and generation modules to address these objectives.
> #### - Furthermore, existing baseline methods treat RCR as a classification task for the USPTO-Condition datasets. To ensure a fair comparison, we conduct a classification module for prediction and evaluation.
> (to be continued)

---

> ### Author Response · Authors · 2024-11-20
> **Response to Weakness 3-4 from Reviewer od79**
>
> #### **Response to weakness 3 :**
> #### We have updated the experimental results which enable SMILES and graph but disable the corpus (line five in Table. 4).   In Table. 4, **SMILES** refers that we introduce reaction representation encoded by SMILES sequences into our model, and **Graph** refers that we consider reaction representation encoded by Graph network.
>
> #### From the results, we can see that different mono-domain data have different contributions for the entire performance. For the prediction of solvent 1, which is the most challenging task, the model enhanced with SMILES representation (first row) outperforms the models trained solely on graph-based features (second row) and corpus data (third row), achieving 21.8% and 23.0% higher top-1 accuracy, respectively.
> #### Subsequently, we investigate how chemical mono-domain data combination affects model performance compared to individual types of data (fourth row to sixth row). By incorporating a corpus into the model already trained with SMILES representations, we achieve a **16.9%** improvement in solvent 1 top-1 prediction accuracy. Similarly, integrating graph features into the SMILES-based model results in a **5.0%** improvement in solvent 1 top-1 accuracy. The effectiveness of incorporating additional corpus data and SMILES representations can be attributed to the LLM’s pre-training on extensive SMILES sequences and reaction data, which equips it with a more comprehensive understanding of chemical reactions and enhances its performance on RCR tasks.
>
> |SMILES | Graph | Corpus | Catalyst (top1/top3/top5) |Solvent1 (top1/top3/top5) | Solvent 2 (top1/top3/top5) | Reagent 1 top1/top3/top5) | Reagent 2 (top1/top3/top5) |
> | ----- | ------ | ------ | ------ | ----- | ------ | ------ | ------ |
> | &#10004; | -- | -- | 90.3 / 97.5 / 98.7 | 37.1 / 64.5 / 75.7 | 80.8 / 92.9 / 96.8 | 37.1 / 63.5 / 74.7 | 73.7 / 89.9 / 94.1 |
>  | -- | &#10004; | -- |  87.1 / 93.3 / 95.5 | 15.3 / 40.5 / 58.2|  80.7 / 91.9 / 95.5| 34.6 / 56.8 / 67.5 | 75.4 / 86.6 / 90.6|
> | -- | -- | &#10004; | 87.1 / 87.4  / 87.8  | 14.1 / 26.1 / 44.9 | 80.7 / 88.1 / 92.0 | 26.0 / 32.1 / 37.3 | 75.1 / 76.6 / 77.9 |
> | &#10004; | -- | &#10004; | 92.6 / 98.5 / 99.3| 54.0  / 76.0 / 84.4 | 81.8 / 94.7 / 97.6 | 52.8 / 75.4 / 83.3 | 78.6 / 93.1 / 96.1 |
> | &#10004;     | &#10004; | -- | 91.3 / 98.1 / 99.1| 42.1 / 68.8 / 79.4 | 80.1 / 93.5 / 97.1 | 45.2 / 70.4 / 79.9 |76.7 / 91.4 / 95.1 |
> |&#10004; | &#10004;|&#10004; | 92.7 / 98.6 / 99.2 | 54.6 / 76.4 / 84.8 | 81.8 / 94.8 / 97.6| 53.4 / 75.8 / 83.9| 78.7 / 93.2 / 96.2|
>
> #### **Response to weakness 4 :**
> #### Thanks for your attention. We add a detailed description of the process of constructing the question-answering datasets in Section 2.1.
>
> #### Toward reaction condition recommendation task in chemical synthesis, we design a tailored instruction prompt system for better cross-modality alignment and instruction tuning (Figure. 2). Instead of traditional instruction prompts for instruction tuning (Figure. 2(a)), we introduce augmented corpus and cross-modalities tokens into constructing instruction prompts (Figure. 2(b)). In particular, each data entry includes three parts: role identification, input instructions, and output answers. ‘Role identification’ (‘Human’) aims to help LLMs identify the chemist (quizzer) and assistant (answerer). By establishing roles, LLMs are encouraged to generate responses that align with the expertise expected in the chemistry domain. The next component ‘input instructions’ is designed to define the user input, allowing LLMs to leverage the given information to generate reasonable responses. Specifically, for a given reaction, we retrieve and collect a corpus—a paragraph contextually similar to the reaction—and populate the <Corpus> placeholder with this information. The reaction is then translated into its corresponding SMILES representation, which is inserted into the <Reaction SMILES> placeholder. Additionally, we design two hint placeholders, <Reaction> and <Graph>, to embed reaction-specific tokens and graph-based representations, respectively. Finally, ’output answers’ specifies the desired format for the response, which is defined directly through a natural language instruction. The expected answer for each question is the combination of chemical conditions, such as ‘Cl.ClCCl’. It is important to note that, to maintain the diversity of instruction datasets, we randomly generate 2,000 question templates using GPT-4 for each pair-wised Q&A. The correctness of all question templates generated by GPT-4 is verified through manual review.

---

> ### Author Response · Authors · 2024-11-20
> **Response to Questions 1-4 from Reviewer od79**
>
> #### Firstly, we want to kindly thank reviewer od79 for the detailed and helpful review. We carefully list our responses to your questions (according to the organized points):
>
> ####  **1. Response to question 1:**
> #### LLMs, through pre-training, have been proven to be endowed with mathematical capabilities [1]. Also, previous work [2-3] has illustrated that to some extent, LLMs could solve chemistry problems but they performed worse than traditional baseline models. Consequently, we suppose that LLMs may have been trained with reaction data. Further, we hypothesize that after being fine-tuned with instruction-based datasets, LLMs can develop enhanced capabilities for chemical reasoning and predictive tasks.
> #### [1] Ahn J, Verma R, Lou R, et al. Large language models for mathematical reasoning: Progresses and challenges[J]. arXiv preprint arXiv:2402.00157, 2024.
> #### [2] Guo T, Nan B, Liang Z, et al. What can large language models do in chemistry? a comprehensive benchmark on eight tasks[J]. Advances in Neural Information Processing Systems, 2023, 36: 59662-59688.
> #### [3] White A D. The future of chemistry is language[J]. Nature Reviews Chemistry, 2023, 7(7): 457-458.
>
> ####  **2. Response to question 2:**
> #### Sure, we had added the citation, which can be seen in the Introduction Section. We propose the Perceiver ~\cite{jaegle2021perceiver} module for modality alignment, which utilizes latent queries to align graphs and SMILES tokens with text-related tokens.
>
> ####  **3. Response to question 3:**
> #### - On the one hand, we illustrate the difference between USPTO-Condition and USPTO_500MT_Condition  datasets by showing the data format of reaction conditions in the ‘Condition illustration’ column in the Table. 1. For the USPTO-Condition dataset, each reaction entry contains five condition labels, including one catalyst ([Zn]),  two solvents (C1CCOC1, O), two reagents (CO, [Cl].[NH4+]), and an additional ``none'' category. Additionally, each data entry in the USPTO_500MT_Condition dataset exclusively includes non-split reagents, represented using dot separators.
> #### - On the other hand, the USPTO-Condition dataset differs from the USPTO_500MT_Condition  dataset in data size, data sparsity, data source, and data distribution. First, we demonstrate the data volume of the USPTO-Condition and USPTO_500MT_Condition  datasets in the Table. 7. The USPTO-Condition dataset is approximately six times larger than the USPTO_500MT_Condition dataset, containing 683,410 entries compared to 109,016. Second, we do a data sparsity analysis of the two datasets shown in Table. 8. From the table, we can see that some conditions, such as ‘Catalyst’, ‘Solvent 2’, and ‘Reagent 2’ show a high extent of sparsity, with a non-empty density of fewer than 30%. In contrast, for the USPTO_500MT_Condition dataset, as it only covers the condition of non-split reagents, all of the reaction entries have their corresponding non-empty condition label. Third, we do a survey on the reaction condition distribution, which is demonstrated in Figure. 5. From the Figure. 5, we can see that there is a significant disparity between the most common condition categories in the two datasets. Furthermore, to prevent data leakage during the generalization experiment, we exclude reactions from the USPTO-Condition training set that overlap with the USPTO_500MT_Condition test set. Thus, we conclude that the two datasets are out-of-distribution.
>
> |Dataset| Sample of conditions | Predictive type | Training set |
> |----- | ----- | ------ | ------ |
> |USPTO-Condition | [Zn],C1CCOC1,O,CO,[Cl-].[NH4+]  | classification | 546,728 |
> |USPTO\_500MT\_Condition | CO.[Na+].CC(=O)O.[BH3-]C\#N   | generation | 88,410 |
>
> ####  **4. Response to question 4:**
> #### Thanks for your suggestions. We have revised the Figure 1. Please check our revised version and we use different boxes to distinguish the data between models.
> (to be continued)

---

> ### Author Response · Authors · 2024-11-20
> **Response to Questions 5-6 from Reviewer od79**
>
> #### Secondly, we want to discuss the left questions,
> #### **5. Response to question 5:**
> ####  Thank you for your suggestions. We use ‘sample of conditions’ for better understanding in our paper. In the Table. 1, we illustrate the difference between USPTO-Condition and USPTO_500MT_Condition datasets by showing the data format of reaction conditions in the sample column. For the USPTO-Condition dataset, each reaction entry contains five condition labels, including one catalyst ([Zn]),  two solvents (C1CCOC1, O), two reagents (CO, [Cl].[NH4+]), and an additional ``none'' category. Additionally, each data entry in the USPTO_500MT_Condition dataset includes only non-split reagents represented with dot separators.
>
> |Dataset| Sample of conditions | Predictive type | Training set |
> |----- | ----- | ------ | ------ |
> |USPTO-Condition | [Zn],C1CCOC1,O,CO,[Cl-].[NH4+]  | classification | 546,728 |
> |USPTO\_500MT\_Condition | CO.[Na+].CC(=O)O.[BH3-]C\#N   | generation | 88,410 |
> #### **6. Response to question 6:**
> #### As we know full-parameter instruction fine-tuning has emerged as an indispensable ingredient in the development of LLMs. However, a notable concern with this fine-tuning is "catastrophic forgetting", where models may lose essential skills. Thus, we decide not to update the parameters of all models, and train only the linear layers including **modality projection** and **condition prediction** modules. The trainable parameters constitute approximately **0.3** billion out of the total **7** billion parameters. Restricting training to the projector modules significantly reduces computational and memory requirements, thereby making the training process faster and more efficient. The training process is conducted with a batch size of 16 for fewer than 6 epochs over 48 hours, utilizing a GPU configuration of 2×48 GB NVIDIA A6000 GPUs. The inference process is highly efficient and can be performed using a single 80 GB NVIDIA A800 GPU.
>
> #### Let us know if anything is missing or confusing. We are happy to further discuss and keep improving our manuscript!

---

> > ### Comment · Reviewer_od79 · 2024-11-25
> >
> > Thank you for the detailed response. While some concerns have been addressed, the following points remain:
> >
> > 1. The introduction requires significant revision, particularly by incorporating the motivation. In Lines 61-65, between "there is currently no ..." and "it is imperative to develop," there should be an explanation addressing *why* this work is necessary.
> >
> > 2. The introduction of the "classification module" and "generation module" appears somewhat arbitrary. Although the authors have provided justification in the rebuttal, it would strengthen the paper to incorporate these explanations directly into the text.
> >
> > 3. In Line 82, there is no citation provided for "perceiver," as previously committed by the authors. This omission needs to be addressed.
> >
> > 4. The statement, *"Thus, here, we suppose that pre-trained with massive chemical reaction data including molecular simplified molecular-input line-entry systems (SMILES) in natural language, LLMs are endowed with fundamental chemical knowledge through text-to-text generation,"* is interesting. However, do the authors believe that this pretraining provides sufficient knowledge for RCR tasks? What additional knowledge, if any, needs to be incorporated into pretrained LLMs? In Lines 305-307, the authors state, *"During the training process, we fix the weight parameters of GCN, reaction encoder, and LLaMA-2. The modality projection and condition prediction layer is trainable."* This suggests no new knowledge is being added to the pretrained LLM (LLaMA-2 in this case). The new modules focus solely on modality alignment and leverage existing knowledge for prediction. Why, then, does this approach achieve such strong performance? Could comparable performance be achieved without additional training, e.g., through in-context learning or prompting, given that the LLM's parameters remain unchanged?

---

> ### Author Response · Authors · 2024-11-26
> **Response to reviewer od79 (Q1-3)**
>
> Dear reviewer od79,
>
> Thank you for taking the time to review our submission and provide your feedback. We appreciate the effort involved in the review process. We mainly revise the introduction part of our manuscript, and we hope the revised contents will make you satisfied. In the meantime, we carefully list our responses to your concerns. Thanks again for your feedback!
>
> 1. We add motivation and explanations to address why our work is necessary in the Introduction (revised version line 44-66, 76-80). Here are the detailed revisions:
> There are various types of data in the field of chemistry, including simplified molecular-input line-entry system (SMILES) (Weininger et al., 1989), graphs, and textual corpus of reaction (Schlichtkrull et al., 2018), which encompasses the descriptions of reaction processes, and reaction mechanisms. Traditional methods tackling the reaction condition recommendation (RCR) task typically rely on sequence-based SMILES data for end-to-end training (Gao et al., 2018; Schwaller et al., 2019; Andronov et al., 2023). However, training exclusively on sequence-based SMILES representations may hinder the model’s ability to capture the difference between similar reactions, as the feature distances encoded by transformers may be too close in the representation space. The capability to encode different reactions is critical for prediction, as even minor variations in a substrate’s functional group can result in fundamentally different reaction conditions. Therefore, it is necessary
> to incorporate additional information into reaction representations for RCR tasks. Given that the textual corpus contains chemical knowledge, which is invaluable for a comprehensive understanding of reactions, we aim to leverage cross-modality data to predict reaction conditions precisely.
> Nowadays, the emergence of generative large language models (LLMs), typified by GPT-4, has sparked significant interest in the field of AI for chemistry (Baum et al., 2021; Achiam et al., 2023; Boiko et al., 2023; Guo et al., 2023; M. Bran et al., 2024). Large multimodal models (LMMs) have demonstrated remarkable predictive capabilities in integrating modalities such as vision, text, and speech (Li et al., 2023; Zhu et al., 2024; Liu et al., 2024a). Therefore, we hypothesize that LMMs
> endowed with LLMs’ foundational capabilities in chemistry can deal with various modalities of chemical data, thereby enhancing the predictive performance in chemical tasks. However, it presents a significant challenge in designing modules to integrate various modalities effectively. Hence, it is imperative to develop an effective prediction model that can incorporate different chemical data into LLMs to achieve a more comprehensive understanding of reaction processes, facilitating the task of chemical reaction condition recommendation.
> 2. We have incorporated these explanations directly into the manuscript text. (revised version: line 188-195)
> 3. You are correct! Seen line 83.
> (to be continued)

---

> ### Author Response · Authors · 2024-11-26
> **Response to reviewer od79 (Q4)**
>
> Dear Reviewer od79,
>
> **Question 4**
>
> (1) We think that pretraining provides foundational chemical knowledge for RCR tasks, as it can grasp the chemical task's objectives, and achieve 30%-40% accuracy performance on benchmark datasets (seen in [1]). Also, we also recognize that incorporating additional domain-specific knowledge, particularly related to reaction mechanism reasoning, may be necessary to enhance performance further.  To evaluate this, we tested GPT-4o and O1 by prompting them to recommend or generate reagents for specific reactions. Below are the question prompts and their corresponding responses:
> ```
> Given a reaction, reactant molecules are O=C(Cl)c1cc(C(F)(F)F)cc(C(F)(F)F)c1.Cc1cccc(C)c1NC(=O)CN1CCN
> ([C@@H]2CCN[C@@H](Cc3ccc(Cl)c(Cl)c3)C2)CC1, product molecule is Cc1cccc(C)c1NC(=O)CN1CCN([C@@H]2CCN(C(
> =O)c3cc(C(F)(F)F)cc(C(F)(F)F)c3)[C@@H](Cc3ccc(Cl)c(Cl)c3)C2)CC1, What would you suggest as potential reagents for this particular chemical reaction?
> ```
> **GPT-4o**: Acid chloride (e.g. benzoyl chloride):
>
> **O1**: Suggested Reagents:
> To facilitate this complex transformation, additional reagents or conditions may be necessary:
> 1. Acidic Conditions:
> - Trifluoroacetic acid (TFA) or acetic acid can promote enolization and stabilize cationic intermediates.
> - Purpose: Protonate the ketone to increase its electrophilicity and stabilize intermediates.
> 2. Solvent:
> - Acetonitrile (MeCN): /Acid chloride (e.g. benzoyl chloride)
>
> We observed that O1 provides more specific and reliable responses compared to GPT-4o. This can be attributed to O1 being enhanced with high-quality reasoning datasets, enabling it to perform more robust reasoning. Based on this observation, we hypothesize that incorporating additional domain-specific knowledge, particularly datasets focused on reaction mechanism reasoning, could further improve performance in the RCR task.
>
> (2) The reason why our model achieves such strong performance is that we construct a high-quality dataset for supervised fine-tuning. This training strategy allows the model to perform strongly by effectively aligning and utilizing existing knowledge to navigate the complexities inherent in reaction condition prediction.  In addition, modality projection is responsible for aligning data from other modalities into the language space. This alignment shortens the distance between well-trained graph representations, reaction SMILES tokens, and language tokens, thereby incorporating more valuable information into the language space. As a result, our model enhances its capability to map multimodal inputs to accurate predictions while retaining foundational chemical knowledge, effectively mitigating the risk of catastrophic forgetting.
>
> In-context learning and prompt engineering techniques have been shown to deliver limited performance in chemistry tasks [1], as they rely solely on the pre-existing knowledge of LLMs. In contrast, Chemma-RC not only leverages the foundational chemical knowledge of LLMs but also integrates cross-modality information into the inherent language space. We suppose that corpus data, SMILES, and graph token representations contain semantic expert knowledge including the interpretability of chemical reactions, mechanism reasoning, etc. The integration of them enriches information significantly enhances the model's ability to learn and perform effectively in RCR tasks.
>
> [1] Guo T, Nan B, Liang Z, et al. What can large language models do in chemistry? a comprehensive benchmark on eight tasks[J]. Advances in Neural Information Processing Systems, 2023, 36: 59662-59688.
>
> Let us know if anything is missing or confusing. We are happy to further discuss and keep improving our manuscript!

---

> > ### Comment · Reviewer_od79 · 2024-11-28
> > **About supervised fine-tuning**
> >
> > Thank you for the clarification.
> >
> > Regarding: "The reason why our model achieves such strong performance is that we construct a high-quality dataset for supervised fine-tuning."
> >
> > Could the author explain this, as I didn't see any description of the supervised fine-tuning of LLaMA-2 throughout the paper or the rebuttal (I though its parameters were frozen)? What is the dataset, what are the details of the supervised fine-tuning (epochs, time, GPU cards, and whether LoRA was used)?

---

> ### Author Response · Authors · 2024-11-29
> **Response to supervised fine-tuning**
>
> Hi! Dear reviewer od79,
>
> I hope my revised introduction part will meet your requirements. **Please let us know if your concerns are addressed and if so, we would be grateful if you are willing to increase your score. We would be happy to discuss further.**
>
> Firstly, we did not use LoRA for supervised fine-tuning in our final model. However, we previously experimented with LoRA, but the performance was suboptimal (see **Predicted Results by Chemma-RC (LoRA)** below). Specifically, we observed that Chemma-RC with LoRA tended to generate repetitive and invalid answers, often producing the same compounds (e.g., "O.O.O"), which negatively impacted the accuracy metric. Even after adjusting the "repetition_penalty", we were unable to achieve pure, precise predictions. We hypothesize that this issue arises because LoRA was applied only to the Q and K layers, which modifies the attention mechanism without significantly enhancing the model's ability to learn new knowledge. Additionally, we applied LoRA to only four layers (excluding the final projection layer), suggesting that the model did not integrate enough domain-specific knowledge to generate accurate predictions aligned with human-defined instructions.
>
> So we decide to update the parameters of the last hidden layers of the LLaMA-2 ---condition prediction module in the manuscript), while freezing the remaining parameters. This approach does not involve strictly supervised fine-tuning for the entire model, allowing us to focus on fine-tuning only the most relevant layers for our task. During the training process, the parameters of condition prediction modules are updated continually. We also construct high-quality Q&A pair-wised datasets for supervised fine-tuning ( detailed in section 3.2.1). A sample case can be seen as follows:
>
> **Question**: Considering a chemical reaction, SMILES is a sequenced-based string used to encode the molecular structure. A chemical reaction includes reactants, conditions, and products. Thus, reactants for this reaction are {CN1C(=O)C(c2ccncc2)(c2cccc(-c3ccccc3)c2)N=C1N.Cl}, SMILES for products of reactions are {CN1C(=O)C(c2cccc(C3CCCCC3)c2)(C2CCNCC2)N=C1N}, then the reaction can be described as {CN1C(=O)C(c2ccncc2)(c2cccc(-c3ccccc3)c2)N=C1N.Cl>>CN1C(=O)C(c2cccc(C3CCCCC3)c2)(C2CCNCC2)N=C1N.Cl}, What would you suggest as potential catalysts for this particular chemical reaction?
>
> **Answer**: O=[Pt]=O
>
> **Predicted results by Chemma-RC**: O=[Pt]=O
>
> **Predicted results by Chemm-RC (LoRA)**: O.O.O.[Na+].[OH-]
>
> In our experiment part (section 4.2 line 312), we only update 0.3 billion parameters out of the total 7 billion parameters. This makes the training more efficient. Also, the training process is conducted with a batch size of 16 for fewer than 6 epochs over 48 hours, utilizing a GPU configuration of 2×48 GB NVIDIA A6000 GPUs. The inference process is highly efficient and can be performed using a single 80 GB NVIDIA A800 GPU.
>
> Thanks again for your attention. Let me know if you have any concerns.

---

> > ### Comment · Reviewer_od79 · 2024-11-29
> >
> > Thanks for the detailed response. I have raised my score.

---

> ### Author Response · Authors · 2024-11-30
> **Repsonse to Reviewer od79**
>
> Hi, Dear reviewer od79,
>
> We highly appreciate your support and helpful suggestions. We believe our discussion during the review did help improve the quality of our manuscript. Thus, We will organically incorporate the discussion into our final revision. Thanks :)

---

### Official Review · Reviewer_GYuz · 2024-11-02

**Soundness:** 3
**Presentation:** 2
**Contribution:** 2
**Rating:** 6
**Confidence:** 3

**Summary:**

The authors present a novel Chemma-RC model designed for chemical reaction recommendation and prediction tasks. This model is trained on multiple data representations, including SMILES notations, reaction graphs, and chemical reaction recommendation corpora. Additionally, the authors introduce an instruction dataset comprising data from various sources, which has been augmented with textual templates. They further demonstrate that the Chemma-RC model can be easily fine-tuned on out-of-distribution data, enabling it to generalize to new types of chemical reactions.

**Strengths:**

The application of language models to chemical reaction planning represents a highly valuable direction for research. While the incorporation of additional data modalities into language models is not entirely novel, it is still an important approach that has the potential to enhance current methodologies by utilizing a broader range of available data. Moreover, the demonstrated ability of the Chemma-RC model to handle out-of-distribution data adds further significance to this work.  Experimental results and ablation study sections are comprehensive.

**Weaknesses:**

While the paper is generally well-written, certain sections lack clarity and sufficient justification for specific decisions made in the study. Additional details on these issues can be found in the Question section.

**Questions:**

* What is the specific purpose of introducing the new dataset? Aside from integrating available data through templating, what novel contributions does this dataset offer? According to the paper, templates were generated using the GPT-4 model. How was the correctness of these generated templates verified?
* The purpose of using reaction graph embeddings remains unclear based on the current ablation study. Integrating graph embeddings requires additional effort including language model’s source code modification, yet Table 4 shows only a slight improvement in model performance with their usage. It appears that relying solely on textual corpora and SMILES representations might be sufficient. It may be beneficial to include an additional ablation study demonstrating cases where graph embeddings significantly enhance performance.
* One of the main contributions of the paper is the integration of multiple data representations and sources within a unified paradigm, positioning the proposed model closer to multi-task, multi-domain foundational models rather than single-task models, mainly presented in the Experimental section. To strengthen the significance of the results, it would be beneficial to include comparisons with current state-of-the-art chemical language models that address reaction planning tasks in their scopes, such as [a, b].

a. BioT5+: Towards Generalized Biological Understanding with IUPAC Integration and Multi-task Tuning, 2024

b. nach0: multimodal natural and chemical languages foundation model, 2024

---

> ### Author Response · Authors · 2024-11-20
> **Response to Questions 1 from Reviewer GYuz**
>
> #### Dear reviewer GYuz,
> #### Thanks for your supportive comments and time to review our paper! Here is our response to your review.
>
> #### **Response to question 1:**
> #### (1) Thanks for your questions. The primary purpose of introducing the new dataset is to facilitate instruction fine-tuning. Recognizing that instruction tuning is crucial for enhancing the performance of large language models (LLMs), we construct this dataset by converting existing benchmark data into instruction-based formats. This process is aimed at improving the efficiency of the training process. By focusing on instruction fine-tuning, we enhance the model's capacity to understand nuanced input instructions and generate accurate responses to corresponding questions. Ultimately, this enables the model to deliver more precise and contextually relevant outputs.
> #### (2-3) Toward reaction condition recommendation task in chemical synthesis, we design a tailored instruction prompt system for better cross-modality alignment and instruction tuning (Figure. 2). Instead of traditional instruction prompts for instruction tuning (Figure. 2(a)), we introduce augmented corpus and cross-modalities tokens into constructing instruction prompts (Figure. 2(b)). In particular, each data entry includes three parts: role identification, input instructions, and output answers. ‘Role identification’ (‘Human’) aims to help LLMs identify the chemist (quizzer) and assistant (answerer). By establishing roles, LLMs are encouraged to generate responses that align with the expertise expected in the chemistry domain. The next component ‘input instructions’ is designed to define the user input, allowing LLMs to leverage the given information to generate reasonable responses. Specifically, for a given reaction, we retrieve and collect a corpus—a paragraph contextually similar to the reaction—and populate the <Corpus> placeholder with this information. The reaction is then translated into its corresponding SMILES representation, which is inserted into the <Reaction SMILES> placeholder. Additionally, we design two hint placeholders, <Reaction> and <Graph>, to embed reaction-specific tokens and graph-based representations, respectively. Finally, ’output answers’ specifies the desired format for the response, which is defined directly through a natural language instruction. The expected answer for each question is the combination of chemical conditions, such as ‘Cl.ClCCl’. It is important to note that, to maintain the diversity of instruction datasets, we randomly generate 2,000 question templates using GPT-4 for each pair-wised Q&A. The correctness of all question templates generated by GPT-4 is verified through manual review.
>
> (to be continued)

---

> ### Author Response · Authors · 2024-11-20
> **Response to Questions 2-3 from Reviewer GYuz**
>
> #### Secondly, we want to further discuss the reaction graph embeddings, and we do some preliminary experimental comparisons.
> #### **Response to question 2:**
>
> #### We have updated the experimental results which enable SMILES and graph but disable the corpus (line five in Table. 4).   In Table. 4, **SMILES** refers that we introduce reaction representation encoded by SMILES sequences into our model, and **Graph** refers that we consider reaction representation encoded by Graph network.
>
> #### From the results, we can see that different mono-domain data have different contributions for the entire performance. For the prediction of solvent 1, which is the most challenging task, the model enhanced with SMILES representation (first row) outperforms the models trained solely on graph-based features (second row) and corpus data (third row), achieving 21.8% and 23.0% higher top-1 accuracy, respectively.
> #### Subsequently, we investigate how chemical mono-domain data combination affects model performance compared to individual types of data (fourth row to sixth row). By incorporating a corpus into the model already trained with SMILES representations, we achieve a **16.9%** improvement in solvent 1 top-1 prediction accuracy. Similarly, integrating graph features into the SMILES-based model results in a **5.0%** improvement in solvent 1 top-1 accuracy. The effectiveness of incorporating additional corpus data and SMILES representations can be attributed to the LLM’s pre-training on extensive SMILES sequences and reaction data, which equips it with a more comprehensive understanding of chemical reactions and enhances its performance on RCR tasks.
>
> |SMILES | Graph | Corpus | Catalyst (top1/top3/top5) |Solvent1 (top1/top3/top5) | Solvent 2 (top1/top3/top5) | Reagent 1 top1/top3/top5) | Reagent 2 (top1/top3/top5) |
> | ----- | ------ | ------ | ------ | ----- | ------ | ------ | ------ |
> | &#10004; | -- | -- | 90.3 / 97.5 / 98.7 | 37.1 / 64.5 / 75.7 | 80.8 / 92.9 / 96.8 | 37.1 / 63.5 / 74.7 | 73.7 / 89.9 / 94.1 |
>  | -- | &#10004; | -- |  87.1 / 93.3 / 95.5 | 15.3 / 40.5 / 58.2|  80.7 / 91.9 / 95.5| 34.6 / 56.8 / 67.5 | 75.4 / 86.6 / 90.6|
> | -- | -- | &#10004; | 87.1 / 87.4  / 87.8  | 14.1 / 26.1 / 44.9 | 80.7 / 88.1 / 92.0 | 26.0 / 32.1 / 37.3 | 75.1 / 76.6 / 77.9 |
> | &#10004; | -- | &#10004; | 92.6 / 98.5 / 99.3| 54.0  / 76.0 / 84.4 | 81.8 / 94.7 / 97.6 | 52.8 / 75.4 / 83.3 | 78.6 / 93.1 / 96.1 |
> | &#10004;     | &#10004; | -- | 91.3 / 98.1 / 99.1| 42.1 / 68.8 / 79.4 | 80.1 / 93.5 / 97.1 | 45.2 / 70.4 / 79.9 |76.7 / 91.4 / 95.1 |
> |&#10004; | &#10004;|&#10004; | 92.7 / 98.6 / 99.2 | 54.6 / 76.4 / 84.8 | 81.8 / 94.8 / 97.6| 53.4 / 75.8 / 83.9| 78.7 / 93.2 / 96.2|
>
> #### **Response to question 3:**
> #### Thanks for your suggestions. We include comparisons with current state-of-the-art chemical language models on USPTO-500MT-Condition datasets, Here are our statements:
> #### From the results, we can see that Chemma-RC demonstrates the most favorable performance on the USPTO-500MT-Condition dataset, where achieves **25.9%** top-1 accuracy when compared with other baseline methods such as Reagent Transformer (17.5%), Reaction GCNN (16.1%), nach0 (13.1%).  We did not include a comparison with BioT5+ because it is specifically designed for molecular understanding, and the instruction datasets used for its training differ from those employed in our study.
>
> |Model | Top-1 accuracy | Top-3 accuracy  | Top-5 accuracy  | Top-10 accuracy  |
> | ----- | ------ | ------ | ------ | ----- |
> | Reagent Transformer | 17.5 | 27.5 | 31.6  | 35.6|
> | Reaction GCNN | 16.1 | 27.5 | 33.0  | 40.2 |
> | Parrot | 13.8 | 25.3 | 31.4 | 37.9 |
> | nach0 | 13.1 | --- | --- | ---|
> | Chemma-RC | 25.9 | 47.2 | 67.8 | 79.2 |
>
> #### Furthermore, we include the other baseline methods for comparison on USPTO-Condition datasets. A detailed explanation of each method and experiment settings can be seen in the revised paper Section 4.4.2 and Appendix. D. In a word, Chemma-RC significantly outperforms all baseline methods across both RS and TS settings. Notably, it achieves a **Top-1 (RS)** accuracy of **72.3%**, which is substantially higher than the second-best approach, TextReact (gr), at 47.2%.
>
> |Model | Top-1 (RS) | Top-3 (RS) | Top-10 (RS)  | Top-15 (RS) | Top-1 (TS) | Top-3 (TS) | Top-10 (TS)  | Top-15 (TS) |
> | ----- | ------ | ------ | ------ | ----- | ----- | ------ | ------ | ------ |
> | rxnfp LSTM | 20.5 | 30.7 | 41.7 | 45.3 | 15.2 | 26.2 | 40.7 | 45.4 |
> | rxnfp retrieval | 27.2 | 37.5 | 47.9 | 51.1 | 7.8  | 15.2 | 27.3 | 31.5 |
> | Transformer | 30.0 | 43.8 | 56.7 | 60.5 | 18.7 | 31.8 | 47.6 | 52.7  |
> | ChemBERTa | 30.3 | 44.7 | 58.0 | 62.0 | 18.7 | 31.9 | 47.6 | 52.8 |
> |TextReact (gr) | 47.2 | 59.9 | 65.0 | 71.4 | 36.3 | 50.4 | 56.2 | 63.8 |
> | Chemma-RC |  72.3 | 87.8| 92.4 | 96.5| 69.6| 86.7 | 91.7| 96.2|

---

> > ### Comment · Reviewer_GYuz · 2024-11-26
> >
> > I sincerely thank the authors for addressing my questions and requests in detail. I hope the additional experimental results, particularly the description of dataset construction, will be included in the final version of the manuscript. Although the results did not fully convince me of the necessity of the graph modality, I still find the quality and novelty of this work sufficient and have decided to maintain my score.

---

### Official Review · Reviewer_3FKb · 2024-11-03

**Soundness:** 2
**Presentation:** 3
**Contribution:** 2
**Rating:** 5
**Confidence:** 5

**Summary:**

The paper titled “Text-Augmented Multimodal Large Language Models for Chemical Reaction Condition Recommendation” introduces Chemma-RC, a novel multimodal large language model designed for chemical reaction condition recommendation (RCR). It leverages SMILES, reaction graphs, and textual corpus to improve the accuracy of RCR predictions and builds a 1.2 million pairwise Q&A dataset to train the model. Experimental results on USPTO benchmark datasets indicate that Chemma-RC achieves strong performance in both in-domain and out-of-domain settings, with a proposed Perceiver module for modality alignment.

**Strengths:**

1. Large-scale training data: Chemma-RC utilizes a dataset comprising 1.2 million pairs of question-and-answer instructions during training. The substantial size of this dataset may contribute to improved learning efficacy and accuracy of the model, enhancing its performance on high-throughput experimental data.

2. Generalization capability: The paper indicates that Chemma-RC performs well on out-of-domain (OOD) and high-throughput experimentation (HTE) datasets, demonstrating its robustness and adaptability across different data distributions.

**Weaknesses:**

1. While the paper presents Chemma-RC as a novel multimodal model, the experimental results lack sufficient evidence that its core techniques—such as modality alignment and instruction tuning—offer a distinct advantage over simpler models like T-Rex and TextReact. To address this, I recommend that the authors include specific comparisons to highlight the unique contributions of Chemma-RC. For example, broadening the set of baseline models beyond T-Rex and TextReact could provide a more comprehensive view of Chemma-RC’s competitive edge.

2. While the manuscript addresses certain aspects of generalization (e.g., OOD and HTE results), it would benefit from additional evaluations to comprehensively demonstrate Chemma-RC’s versatility in chemical reaction modeling. For instance, models like TextReact illustrate their flexibility by performing multiple tasks, including reaction condition prediction and retrosynthesis. To parallel these capabilities, I suggest that the authors add experiments to assess Chemma-RC on diverse predictive tasks, such as reaction condition recommendation (RCR) using the USPTO dataset, as in TextReact. Additionally, it would be valuable to report how Chemma-RC performs under different dataset splits, such as random split (RS) and time split (TS), to further highlight its robustness across varied conditions.

3. The construction and training of Chemma-RC, as a complex multimodal model, may demand substantial computational resources and time, potentially creating challenges for research teams with limited access to high-performance computing. To make the model more accessible, I suggest the authors provide details on the computational requirements (e.g., training time, hardware specifications) and consider exploring ways to reduce these demands, such as model simplifications or alternative training strategies. This would offer clearer guidance for teams looking to replicate or adapt Chemma-RC within resource constraints.

4. While integrating multiple modalities indeed enhances the model's capabilities, it also increases its complexity, potentially leading to extended training times and additional debugging challenges. To assist others in implementing or building upon this work, I suggest that the authors discuss any specific challenges they encountered during the model’s training and debugging process due to its complexity. Insights into how these issues were addressed, including strategies or tools used to streamline debugging, would be valuable additions to the community and provide practical guidance for future work.

5. The model’s dependence on a large volume of high-quality multimodal data may limit its applicability in fields with limited data availability. I recommend that the authors discuss how Chemma-RC might perform with smaller datasets and explore potential strategies for adapting it to data-scarce domains. For example, they could consider approaches such as transfer learning, data augmentation, or semi-supervised learning to enhance the model's applicability where data is less abundant. Including these insights would make the model’s utility more broadly applicable to real-world scenarios.

6. Given the model's complexity, there is an inherent risk of overfitting, particularly in cases of limited or noisy training data. I suggest that the authors provide details on any regularization techniques or strategies they employed to mitigate overfitting. Additionally, discussing how they validated the model’s performance on unseen data—such as through cross-validation or using an independent test set—would help clarify the model’s robustness and reliability. These insights would be valuable for understanding the model's generalization capabilities in varied data environments.

**Questions:**

1. What are the results of the reaction condition recommendation (RCR) when using the USPTO dataset employed by the TextReact model, and how do the results differ when the dataset is partitioned according to RS (random split) and TS (time split)?

2. In Abstract: The term "unified reaction representation" might need clarification. It would be helpful to specify how the SMILES, reaction graphs, and textual corpus are integrated into this unified representation.

3. In Introduction (Lines 38-39): The problem of "low sparsity of chemical data" is mentioned but could be better explained. It’s unclear if this refers to limited diversity in reaction data or the low availability of high-quality reaction data.

4. Figure 1 - Model architecture description: The diagram of Chemma-RC's architecture (Figure 1) is complex, and the explanation in the text doesn’t fully clarify how each component (e.g., modality projection and condition prediction) interacts within the model. A step-by-step walkthrough could enhance understanding.

5. In section 3.2.1 - Text-Augmented Instruction Prompts: The description of how prompts are constructed might benefit from more examples, especially regarding "question templates" and their variety. It is unclear how "2,000 question templates" differ in structure or content and how these templates influence model training.

6. In section 4.3 - Performance Comparison (Tables 2 and 3): The term "strict matching policy" for top-k accuracy might need clarification. It’s unclear what criteria are used for strict matching—whether it’s an exact match of predicted and true labels or some other criteria.

7. Ablation Study (Section 4.4): While this section reports performance changes with various data types (SMILES, Graph, Corpus), it would be beneficial to clarify how the model interprets these individually. For example, what specific information from each modality improves the prediction?

8. In section 4.6 - Zero-Shot Prediction on High-Throughput Experimentation: The phrase "top-1 partial match accuracy" in the results table needs explanation. It is unclear what is considered a partial match versus a complete match, particularly for reagent and solvent prediction.

9. In Conclusion and Limitations: The paper mentions plans to "optimize data representation with full fine-tuning training strategies" but doesn’t clarify why full fine-tuning was not initially used. It would be helpful to understand the limitations of the current training strategy and how it impacts model performance.

10. In Appendix B - Data Description: In the dataset descriptions, some technical terms like "sparsity analysis" might be confusing. A brief explanation of "non-empty count" and "density" could aid in understanding the data distribution.

---

> ### Author Response · Authors · 2024-11-20
> **Response to Weakness 1-3 from Reviewer 3FKb**
>
> #### Dear reviewer 3FKb,
> #### We want to extend our heartfelt thanks for your comments on our manuscript and the detailed suggestions. Here are the responses to the weaknesses and questions:
>
> #### **Response to weakness 1:**
> #### Thank you so much for suggesting the related work.  We have broadened several sets of baseline models to illustrate the feasibility of Chemma-RC, including nach0, transformer-based models, etc. T-Rex you mentioned is not our scope as it focuses on retrosynthesis tasks. We include comparisons with current state-of-the-art chemical language models on the USPTO_500MT_Condition dataset (shown in Table below). Results demonstrate that Chemma-RC has the most favorable performance, where achieves **25.9%** top-1 accuracy when compared with other baseline methods such as Reagent Transformer (17.5%), Reaction GCNN (16.1%), nach0 (13.1%).
>
> |Model | Top-1 accuracy | Top-3 accuracy  | Top-5 accuracy  | Top-10 accuracy  |
> | ----- | ------ | ------ | ------ | ----- |
> | Reagent Transformer | 17.5 | 27.5 | 31.6  | 35.6|
> | Reaction GCNN | 16.1 | 27.5 | 33.0  | 40.2 |
> | Parrot | 13.8 | 25.3 | 31.4 | 37.9 |
> | nach0 | 13.1 | --- | --- | ---|
> | Chemma-RC | 25.9 | 47.2 | 67.8 | 79.2 |
>
> #### [1] nach0: multimodal natural and chemical languages foundation model, 2024
>
> #### **Response to weakness 2:**
> #### Furthermore, we include the other baseline methods for comparison on USPTO-Condition datasets. A detailed explanation of each method and experiment settings can be seen in the revised paper Section 4.4.2 and Appendix. D. From the results, we can see that the performance of other baseline models such as rxnfp LSTM, rxnfp retrieval, Transformer, and ChemBERTa shows moderate success. However, these models consistently deliver lower accuracy rates compared to TextReact (gr) and Chemma-RC. Chemma-RC significantly outperforms all baseline methods across both RS and TS settings. Notably, it achieves a **Top-1 (RS)** accuracy of **72.3%**, which is substantially higher than the second-best approach, TextReact (gr), at 47.2%.
> |Model | Top-1 (RS) | Top-3 (RS) | Top-10 (RS)  | Top-15 (RS) | Top-1 (TS) | Top-3 (TS) | Top-10 (TS)  | Top-15 (TS) |
> | ----- | ------ | ------ | ------ | ----- | ----- | ------ | ------ | ------ |
> | rxnfp LSTM | 20.5 | 30.7 | 41.7 | 45.3 | 15.2 | 26.2 | 40.7 | 45.4 |
> | rxnfp retrieval | 27.2 | 37.5 | 47.9 | 51.1 | 7.8  | 15.2 | 27.3 | 31.5 |
> | Transformer | 30.0 | 43.8 | 56.7 | 60.5 | 18.7 | 31.8 | 47.6 | 52.7  |
> | ChemBERTa | 30.3 | 44.7 | 58.0 | 62.0 | 18.7 | 31.9 | 47.6 | 52.8 |
> |TextReact (gr) | 47.2 | 59.9 | 65.0 | 71.4 | 36.3 | 50.4 | 56.2 | 63.8 |
> | Chemma-RC |  72.3 | 87.8| 92.4 | 96.5| 69.6| 86.7 | 91.7| 96.2|
>
> #### **Response to weakness 3:**
> ####  Thanks for your interest!  We acknowledge that constructing and training a complex multimodal model like Chemma-RC can be resource-intensive. Here are some details and considerations we're addressing to enhance accessibility, and detailed statements can be found in Appendix. A.  To ensure the accessibility of our models, we decide not to update the parameters of all models, and train only the linear layers including **modality projection** and **condition prediction** modules. The trainable parameters constitute approximately **0.3 billion** out of the total **7 billion** parameters. Restricting training to the projector modules significantly reduces computational and memory requirements, thereby making the training process faster and more efficient. The training process is conducted with a batch size of 16 for fewer than 6 epochs over 48 hours, utilizing a GPU configuration of 2×48 GB NVIDIA A6000 GPUs. The inference process is highly efficient and can be performed using a single 80 GB NVIDIA A800 GPU.
> (to be continued)

---

> ### Author Response · Authors · 2024-11-20
> **Response to Weakness 4- 5 from Reviewer 3FKb**
>
> #### Dear reviewer 3FKb,
> #### **Response to weakness 4:**
> #### Thank you for highlighting the challenges associated with integrating multiple modalities. We agree that the complexity of Chemma-RC can introduce extended training times and debugging challenges. Here are some insights into our experience and strategies employed:
> #### &ensp; - **Use of Advanced Tools:** Tools such as TensorBoard and custom logging scripts are utilized to continuously monitor model performance and identify anomalies early in the training process.
> #### &ensp; - **Automated Testing Pipelines:** Implementing automated tests for each modality independently facilitates quicker identification of integration issues.
>
> #### Here are some best practices and recommendations:
> #### &ensp; - **Incremental Integration:** Gradually integrating modalities allows us to confirm functionality at each stage before moving forward, reducing the scope of potential issues.
> #### &ensp; - **Modular Design:** Designing the model with modularity in mind significantly simplifies testing, debugging, and potential future modifications.
>
> #### **Response to weakness 5:**
> #### Thanks for your suggestions. We agree that LLMs limit their applicability in fields with limited data availability. That being said, we evaluate the condition recommendation capabilities of Chemma-RC on High-Throughput Experimentation (HTE) datasets (Section 4.6). First, we ensure that reaction data of imidazole C--H functionalization is excluded from the test set of the USPTO-Condition dataset to prevent data leakage issues. Chemma-RC recommends a ligand under a pre-defined solvent-base combination of conditions. As shown in Figure. 3, we randomly select six cases for performance evaluation. The referenced bases, solvents, and ligands can be found in the reaction formula, which has been annotated by 'B', 'S', and 'L'. In Figure. 3, under the combination of CsOAc and DMAc, Chemma-RC identifies the XPhos ligand, which results in a higher yield. For **15** of the 16 base-solvent combinations, the recommended ligand performs best in terms of the median value of reaction yields, suggesting that Chemma-RC can recommend ligands with higher yields.
>
> #### Furthermore, we also illustrate the flexibility of our models by performing multiple tasks. Specifically,  we evaluate the performance of Chemma-RC for the **reaction reactivity prediction** task. We select the Ni‐catalyzed Suzuki‐Miyaura Couplings reaction for evaluation. This HTE dataset consists of 480 reaction entries across five template reaction types. Reactants, reaction conditions, and products are utilized as inputs to construct an instruction-based Q&A dataset, with the question template designed to predict reaction reactivity by GPT-4. To facilitate few-shot transfer learning, the training data we use are aligned with the baseline method in [1]. Fine-tuning with only 90 reaction entries, the model demonstrates the following performance. The table below reports the accuracy of reaction reactivity prediction.
>
> |Reactions | Chemma-RC| Baseline method|
> | ----- | ------ | ------ |
> | Raection 1 | 0.86 | 0.84 |
> | Raection 2 | 0.82 | 0.88 |
> | Raection 3 | 0.87 | 0.76 |
> | Raection 4 | 0.93 | 0.79 |
> | Raection 5 | 0.83 | 0.86 |
>
> #### From the results, we can see that Chemma-RC demonstrates higher accuracy in several reactions compared to the Multi-Threshold method. Particularly noteworthy is **Reaction 3**, where Chemma-RC achieves an accuracy of **0.87**, substantially outperforming the Multi-Threshold approach at 0.76. These results highlight Chemma-RC's strong generalization performance in reaction reactivity prediction and its potential applicability in data-scarce domains, such as HTE datasets.
> #### Since our work focuses on the RCR task, we will not include these experimental results for reaction reactivity prediction in our revised paper. Here, we want to demonstrate that, Chemma-RC, through few-shot transfer learning, performs effectively even with smaller datasets, showcasing its adaptability to data-scarce domains. In the future, we will focus on exploring and discussing the capabilities of Chemma-RC, such as exploration in an open reaction space.
>
> #### [1] LeSueur A, Tao N, Doyle A, et al. Multi‐Threshold Analysis for Chemical Space Mapping of Ni‐Catalyzed Suzuki‐Miyaura Couplings[J]. European Journal of Organic Chemistry, 2024: e202400428.
>  #### (to be continued)

---

> ### Author Response · Authors · 2024-11-20
> **Response to Questions 1-6 from Reviewer 3FKb**
>
> #### Dear reviewer 3FKb,
> #### We want to extend our heartfelt thanks again for your comments on our manuscript and the detailed suggestions. Here are the responses to the all questions.
>
> #### **Response to question 1:**
> #### As we have discussed in the responses for weakness, we present results when the dataset is partitioned according to RS (random split) and TS (time split). We include the other baseline methods for comparison on USPTO-Condition datasets. A detailed explanation of each method and experiment settings can be seen in the revised paper Section 4.4.2 and Appendix. D. From the results, we can see that the performance of other baseline models such as rxnfp LSTM, rxnfp retrieval, Transformer, and ChemBERTa shows moderate success. However, these models consistently deliver lower accuracy rates compared to TextReact (gr) and Chemma-RC. Chemma-RC significantly outperforms all baseline methods across both RS and TS settings. Notably, it achieves a **Top-1 (RS)** accuracy of **72.3%**, which is substantially higher than the second-best approach, TextReact (gr), at 47.2%.
> |Model | Top-1 (RS) | Top-3 (RS) | Top-10 (RS)  | Top-15 (RS) | Top-1 (TS) | Top-3 (TS) | Top-10 (TS)  | Top-15 (TS) |
> | ----- | ------ | ------ | ------ | ----- | ----- | ------ | ------ | ------ |
> | rxnfp LSTM | 20.5 | 30.7 | 41.7 | 45.3 | 15.2 | 26.2 | 40.7 | 45.4 |
> | rxnfp retrieval | 27.2 | 37.5 | 47.9 | 51.1 | 7.8  | 15.2 | 27.3 | 31.5 |
> | Transformer | 30.0 | 43.8 | 56.7 | 60.5 | 18.7 | 31.8 | 47.6 | 52.7  |
> | ChemBERTa | 30.3 | 44.7 | 58.0 | 62.0 | 18.7 | 31.9 | 47.6 | 52.8 |
> |TextReact (gr) | 47.2 | 59.9 | 65.0 | 71.4 | 36.3 | 50.4 | 56.2 | 63.8 |
> | Chemma-RC |  72.3 | 87.8| 92.4 | 96.5| 69.6| 86.7 | 91.7| 96.2|
>
> #### **Response to question 2:**
> #### Thanks for your advice.
> * ####  Firstly, we revise the abstract as follows: Here, we present Chemma-RC, a text-augmented multimodal LLM that responds to task-specific questions by generating answers about reaction conditions. It learns a unified reaction representation via modality alignment from a corpus of reactions and question prompts, molecular structures in SMILES format, and graphical representations of chemical reactions.
>
> * #### Secondly, to be specific, we will explain how the SMILES, reaction graphs, and textual corpus are integrated into this unified representation. To be specific, given a reaction, we retrieve a relevant corpus—a paragraph containing contextual information that closely resembles the reaction—and populate the <Corpus> placeholder with this data. Next, the reaction is converted into its corresponding SMILES representation, which is then inserted into the <Reaction SMILES> placeholder. Finally, we introduce two additional placeholders, <Reaction> and <Graph>, designed to accommodate the reaction and graph-based representations, respectively. In instruction fine-tuning, all reaction embedding representations are employed as latent tokens to align text-related tokens via modality alignment.
>
> #### **Response to question 3:**
> #### Sorry for the misunderstanding. We update the expression: However, few efforts have been made to solve the problem of reaction condition screening due to the limited diversity of conditions in reaction data.
> #### **Response to question 4:**
> #### Thanks for your suggestions. We have revised the Figure 1. Please check our revised version and we use different boxes to distinguish the data from models.
> #### **Response to question 5:**
> #### Thanks for your advice. We add more examples of the question templates generated by GPT-4, which can be seen as follows and Appendix Table. 8. It is important to note that, we leverage GPT-4 to generate distinct question templates for question prompts, following the policy of **diversity, consistency, and completeness**. The correctness of all question templates generated by GPT-4 is verified through manual review. A comprehensive and diverse set of prompts allows Chemma-RC to better generalize its understanding and be robust to inquiries proposed by chemists.
>
> |Task | Description|
> | ----- | ------ |
> | Solvent prediction | Could you suggest potential solvents that could have been used in the given chemical reaction, taking into consideration their polarity and compatibility with the reactants? |
> | Reagent prediction | Please suggest some possible reagents that could have been used in the following chemical reaction. |
> | Catalyst prediction | Considering the chemical reaction in question, which catalysts could be effective? |
> | Condition prediction (all)| Given the current chemical reaction, what would be the appropriate conditions to consider? |
>
> #### **Response to question 6:**
> #### The term "strict matching policy" denotes a metric that requires an exact match between the predicted and true labels. Under this policy, the predicted results are deemed correct only when the SMILES sequences of the predictions are identical to those of the true labels.
>  (to be continued)

---

> ### Author Response · Authors · 2024-11-20
> **Response to Questions 7-10 from Reviewer 3FKb**
>
> #### **Response to question 7:**
> #### Thanks for this point. We believe that it is very important to discuss the effect of each modality data for the entire performance. We add ablation studies to discuss this, and the experimental results are as follows.
>
> #### From the results, we can see that different mono-domain data have different contributions for the entire performance. For the prediction of solvent 1, which is the most challenging task, the model enhanced with SMILES representation (first row) outperforms the models trained solely on graph-based features (second row) and corpus data (third row), achieving 21.8% and 23.0% higher top-1 accuracy, respectively.
> #### Subsequently, we investigate how chemical mono-domain data combination affects model performance compared to individual types of data (fourth row to sixth row). By incorporating a corpus into the model already trained with SMILES representations, we achieve a **16.9%** improvement in solvent 1 top-1 prediction accuracy. Similarly, integrating graph features into the SMILES-based model results in a **5.0%** improvement in solvent 1 top-1 accuracy. The effectiveness of incorporating additional corpus data and SMILES representations can be attributed to the LLM’s pre-training on extensive SMILES sequences and reaction data, which equips it with a more comprehensive understanding of chemical reactions and enhances its performance on RCR tasks.
>
> |SMILES | Graph | Corpus | Catalyst (top1/top3/top5) |Solvent1 (top1/top3/top5) | Solvent 2 (top1/top3/top5) | Reagent 1 top1/top3/top5) | Reagent 2 (top1/top3/top5) |
> | ----- | ------ | ------ | ------ | ----- | ------ | ------ | ------ |
> | &#10004; | -- | -- | 90.3 / 97.5 / 98.7 | 37.1 / 64.5 / 75.7 | 80.8 / 92.9 / 96.8 | 37.1 / 63.5 / 74.7 | 73.7 / 89.9 / 94.1 |
>  | -- | &#10004; | -- |  87.1 / 93.3 / 95.5 | 15.3 / 40.5 / 58.2|  80.7 / 91.9 / 95.5| 34.6 / 56.8 / 67.5 | 75.4 / 86.6 / 90.6|
> | -- | -- | &#10004; | 87.1 / 87.4  / 87.8  | 14.1 / 26.1 / 44.9 | 80.7 / 88.1 / 92.0 | 26.0 / 32.1 / 37.3 | 75.1 / 76.6 / 77.9 |
> | &#10004; | -- | &#10004; | 92.6 / 98.5 / 99.3| 54.0  / 76.0 / 84.4 | 81.8 / 94.7 / 97.6 | 52.8 / 75.4 / 83.3 | 78.6 / 93.1 / 96.1 |
> | &#10004;     | &#10004; | -- | 91.3 / 98.1 / 99.1| 42.1 / 68.8 / 79.4 | 80.1 / 93.5 / 97.1 | 45.2 / 70.4 / 79.9 |76.7 / 91.4 / 95.1 |
> |&#10004; | &#10004;|&#10004; | 92.7 / 98.6 / 99.2 | 54.6 / 76.4 / 84.8 | 81.8 / 94.8 / 97.6| 53.4 / 75.8 / 83.9| 78.7 / 93.2 / 96.2|
>
> #### **Response to question 8:**
> #### Thanks for your attention. We mentioned "top-1 partial match accuracy" in Section 4.5 Generalization Performance. (: Maybe you refer to this part. To evaluate the generalization capabilities of Chemma-RC, we integrate the two split condition labels with dot separators as the condition ground truth for reagents and solvents prediction, respectively. Then we compare the condition predictions with the ground truth and adopt the metric of partial matched accuracy. Different from the complete matched accuracy that requires perfect matching between predictions and labels, the partial matched accuracy is more suitable to test the generalization capacity, which focuses more on whether the predicted results match a substitutable part of the ground truth. For example, if the predicted result is ‘[Na+].[OH-]’ and the condition label is ‘CO.[Na+].[OH-]’, we consider that the prediction partially matches the ground truth, but not completely. From the evaluation results of the partial matched accuracy, we can conclude that our Chemma-RC can successfully distinguish reagents from the combination of all conditions in a reaction, as it learns the relationships between reaction conditions effectively.
> (to be continued)

---

> ### Author Response · Authors · 2024-11-20
> **Response to Questions 9-10 from Reviewer 3FKb**
>
> #### **Response to question 9:**
> * #### As we know, full-parameter instruction fine-tuning has emerged as an indispensable ingredient in the development of LLMs. However, a notable concern with this fine-tuning is "catastrophic forgetting", where models may lose essential skills. In our paper, a full fine-tuning strategy is not initially used.  The reason is that we aim to preserve the diversity of Chemma-RC’s responses, ensuring it goes beyond merely fitting the data. Subsequently, Chemma-RC can be utilized to explore new conditions within an open reaction space, serving as an alternative to traditional active learning approaches for reaction optimization. Furthermore, we only update the parameters of modality projection and condition prediction modules (Figure. 1). This strategy reduces computational demands and memory usage, thereby accelerating the training process and improving efficiency.
> ####  Limitations of the current training strategy:
> * #### **Scalability**: As instruction fine-tuning requires curated datasets of instruction-response pairs, scaling this process is both time and resource-intensive. This limits the ability to frequently update or expand the model’s capabilities with new and diverse instructions.
> * #### **Overfitting**: There is a risk that the model becomes too tailored to the specific instructions seen during fine-tuning, leading it to overfit these patterns and thereby exhibit reduced flexibility when encountering novel or rephrased instructions.
> * #### **Transferability**: Instruction fine-tuning may improve the model's performance on specific learned tasks but not necessarily on other related tasks unless explicitly trained for those scenarios.
>
> #### **Response to question 10:**
> ####  We are very sorry for the confusion. Here, we add more statements to explain sparsity analysis.  Firstly, in the USPTO-Condition dataset, each chemical context condition (catalyst, solvent1, solvent2, reagent1, and reagent2) will be symbolized as "None" if the reaction does not require this type of reaction condition. To gain deeper insight into the data distribution, we calculate the "non-empty count" and "non-empty density" for each condition, as illustrated in Table 8. Specifically, the "non-empty count" is determined by calculating the entries not marked as "None”, and the "non-empty density" is defined as the ratio of the "non-empty count" to the total size of the dataset.
>
> (to be continued)

---

> ### Author Response · Authors · 2024-11-21
> **Response to Weakness 6 from Reviewer 3FKb**
>
> #### Dear reviewer 3FKb,
> #### We notice that you are very concerned about the training process including training strategies and debugging challenges. Here, we discuss detailed debugging suggestions, and hope will make you satisfied. Furthermore, we will release our up-to-date model and source code to contribute to the chemistry community.
> #### **Response to weakness 6:**
> #### &ensp; 1. Regularization Techniques:
> * #### We incorporate dropout layers at various stages of the model to prevent overfitting by randomly deactivating a fraction of neurons during training.
> * #### We use warm-up strategies for the decay of the learning rate.
>
> #### &ensp; 2. Validation Strategies:
> * ####  Cross-Validation: We employ k-fold cross-validation to evaluate the model's performance across different subsets of the data, ensuring that our results were consistent and not tied to a specific data split.
> * #### Independent Test Set: The model is tested on an independent, unseen dataset to assess its performance in real-world conditions. This approach provides a clear measure of the model's generalization capabilities.
>
> #### &ensp; 3. Monitoring Mechanisms:
> * ####  Early stopping criteria are implemented to halt training once the model's performance on a validation set ceases to improve, preventing unnecessary training that could lead to overfitting.
>
> #### Here are all responses to weaknesses!  Let us know if anything is missing or confusing. We are happy to further discuss and keep improving our manuscript!

---

> > ### Comment · Reviewer_3FKb · 2024-11-27
> >
> > Thanks to the authors for their comprehensive responses.
> >
> > Reading the responses I think many of my concerns have been addressed, but some key questions remain. Specifically, regarding Weakness 1, Question 5, and Question 7, the authors have not fully clarified why incorporating textual information improves the model's accuracy. Additionally, the work focuses on multimodal data but lacks theoretical justification for how different modalities contribute to predicting reaction conditions or overall performance improvements. This conclusion is counterintuitive, as many recent studies have shown that graph- or SMILES-based models often outperform even the most advanced text-only models, such as LLMs. A comparison across different prompts would also provide valuable insights.
> >
> > Additionally, exploring whether using more advanced language models (e.g., GPT-4o, Llama 3.2) or chemistry-specific models (e.g., ChemLLM, ChemGPT) could enhance the framework’s effectiveness would be beneficial.
> >
> > The paper could benefit from a more in-depth analysis of the fusion mechanism, moving beyond simple concatenation and potentially comparing it with alternative fusion strategies.
> >
> > Overall, I appreciate the authors' efforts and have decided to maintain my score.

---

> ### Author Response · Authors · 2024-11-28
> **Response to Reviewer 3FKb**
>
> Thanks for your feedback and for taking the time to review our paper! Could you please provide a list of recent works that outperform advanced text-only models, including LLMs for RCR tasks, so that we can compare these methods for potential improvements in our approach? Recently, many scientists have been striving to explore the potential of LLMs in chemistry, driven by the belief that there exists a possibility to surpass the current graph or SMILES-based methods [1-4]. In addition, the ChemLLM, and ChemGPT you mentioned did not include the RCR tasks. Based on the related work insights, we are the first to design chemical multimodal LLM to facilitate RCR tasks. Our research confirms the potential that LLMs outperform traditional methods. Although further refinement and testing of Chemma-RC's explanations are still required, this also highlights the value of our work and its potential for future exploration. In this paper, we aim to publish these findings promptly to contribute to the broader AI-chemistry community.
>
> [1] Autonomous chemical research with large language models, Nature;
>
> [2] Augmenting large language models with chemistry tools, Nature Machine Intelligence;
>
> [3] The future of chemistry is language, Nature Reviews Chemistry;
>
> [4] ChatMOF: an artificial intelligence system for predicting and generating metal-organic frameworks using large language models, Nature Communications；

---

> ### Author Response · Authors · 2024-11-30
> **Response to Reviewer 3FKb about Weakness 1, Question 5, and Question 7.**
>
> Hi, Dear Reviewer 3FKb,
>
> Thanks for all your questions and concerns. We would like you to reconsider our work (:. **Please let us know if your concerns are addressed and if so, we would be grateful if you are willing to increase your score. We would be happy to discuss further.** We decide to discuss further some key questions you mentioned including Weakness 1, Question 5, and Question 7.
>
> **Weakness 1**
>
> For weakness 1, we added some baseline methods for comparison (see response W1 above). Results demonstrate that Chemma-RC has the most favorable performance. Traditional methods tackling the reaction condition recommendation (RCR) task typically rely on sequence-based SMILES data for end-to-end training (Gao et al., 2018; Schwaller et al., 2019; Andronov et al., 2023). However, training exclusively on sequence-based SMILES representations may hinder the model’s ability to capture the difference between similar reactions, as the feature distances encoded by transformers may be too close in the representation space. The capability to encode different reactions is critical for prediction, as even minor variations in a substrate’s functional group can result in fundamentally different reaction conditions. Therefore, it is necessary to incorporate additional information into reaction representations for RCR tasks. Given that the textual corpus contains chemical knowledge, which is invaluable for a comprehensive understanding of reactions, we aim to leverage cross-modality data to predict reaction conditions precisely.
>
> **Question 5**
>
> Thanks for your questions. We leverage GPT-4 to generate distinct question templates for question prompts and ask  it to follow the policy of diversity, consistency, and completeness. Here are our prompts to generate question prompts:
> ```
> ​​You are a highly capable chemical assistant specializing in addressing reaction condition recommendation tasks. Given the reactants and products of a reaction, your role is to generate suitable conditions for the reaction, including reagents, solvents, and catalysts. Your task is to assist in creating 2,000 question templates for each type of condition recommendation. Please ensure that all generated templates are precise, diverse, consistent, and complete. For instance, if the goal is to recommend reagents for a given reaction, a possible template could be: "What would you suggest as potential reagents for this particular chemical reaction?
> ```
>
> Meanwhile, the correctness of all question templates generated by GPT-4 is verified through manual review to ensure quality and consistency. Additionally, we conducted experiments where the questioning style in the training datasets was intentionally kept consistent. Our experiments have shown that maintaining a consistent questioning style during training strongly impacts the prediction outcomes. Specifically, if the questioning style used during inference differs from the templates used during training, the model is more likely to produce incorrect predictions, significantly reducing its robustness. To enhance the model's robustness to varied questioning styles, it is crucial to maintain diversity in the training corpus.

---

> > ### Author Response · Authors · 2024-11-30
> > **More detailed explanations**
> >
> > [continued]
> > **Question 7**
> >
> > Based on the supplementary experiments, we conclude that both SMILES and corpus representations contribute positively to model performance. Since SMILES representation is a sequence-based expression, previous studies have demonstrated that LLMs are capable of recognizing and processing this format. Therefore, we conclude that incorporating additional domain-specific knowledge, particularly related to reaction mechanism reasoning, may be necessary to enhance performance further. To evaluate this, we tested GPT-4o and O1 by prompting them to recommend or generate reagents for specific reactions. Below are the question prompts and their corresponding responses:
> >
> >
> > ```
> > Given a reaction, reactant molecules are O=C(Cl)c1cc(C(F)(F)F)cc(C(F)(F)F)c1.Cc1cccc(C)c1NC(=O)CN1CCN
> > ([C@@H]2CCN[C@@H](Cc3ccc(Cl)c(Cl)c3)C2)CC1, product molecule is Cc1cccc(C)c1NC(=O)CN1CCN([C@@H]2CCN(C(
> > =O)c3cc(C(F)(F)F)cc(C(F)(F)F)c3)[C@@H](Cc3ccc(Cl)c(Cl)c3)C2)CC1, What would you suggest as potential reagents for this particular chemical reaction?
> > ```
> > **GPT-4o**: Acid chloride (e.g. benzoyl chloride):
> >
> > **O1**: Suggested Reagents:
> > To facilitate this complex transformation, additional reagents or conditions may be necessary:
> > 1. Acidic Conditions:
> > - Trifluoroacetic acid (TFA) or acetic acid can promote enolization and stabilize cationic intermediates.
> > - Purpose: Protonate the ketone to increase its electrophilicity and stabilize intermediates.
> > 2. Solvent:
> > - Acetonitrile (MeCN): /Acid chloride (e.g. benzoyl chloride)
> >
> > We observed that O1 provides more specific and reliable responses compared to GPT-4o. This can be attributed to O1 being enhanced with high-quality reasoning datasets, enabling it to perform more robust reasoning. Based on this observation, we hypothesize that incorporating additional domain-specific knowledge, particularly datasets focused on reaction mechanism reasoning, could further improve performance in the RCR task.
> >
> > **About this conclusion is counterintuitive**
> >
> > The reason why our model achieves such strong performance is that we construct a high-quality dataset for supervised fine-tuning. We decide to update the parameters of the last hidden layers of the LLaMA-2 ---condition prediction module in the manuscript), while freezing the remaining parameters. This approach does not involve strictly supervised fine-tuning for the entire model, allowing us to focus on fine-tuning only the most relevant layers for our task. During the training process, the parameters of condition prediction modules are updated continually. Below is a sample data case for training. This training strategy allows the model to perform strongly by effectively aligning and utilizing existing knowledge to navigate the complexities inherent in reaction condition prediction. In addition, modality projection is responsible for aligning data from other modalities into the language space. This alignment shortens the distance between well-trained graph representations, reaction SMILES tokens, and language tokens, thereby incorporating more valuable information into the language space. As a result, our model enhances its capability to map multimodal inputs to accurate predictions while retaining foundational chemical knowledge, effectively mitigating the risk of catastrophic forgetting.
> >
> > A sample case can be seen as follows:
> >
> > ```
> > Question: Considering a chemical reaction, SMILES is a sequenced-based string used to encode the molecular structure. A chemical reaction includes reactants, conditions, and products. Thus, reactants for this reaction are {CN1C(=O)C(c2ccncc2)(c2cccc(-c3ccccc3)c2)N=C1N.Cl}, SMILES for products of reactions are {CN1C(=O)C(c2cccc(C3CCCCC3)c2)(C2CCNCC2)N=C1N}, then the reaction can be described as {CN1C(=O)C(c2ccncc2)(c2cccc(-c3ccccc3)c2)N=C1N.Cl>>CN1C(=O)C(c2cccc(C3CCCCC3)c2)(C2CCNCC2)N=C1N.Cl}, What would you suggest as potential catalysts for this particular chemical reaction?
> > ```
> > Thanks again!!. We hope our responses can solve your concerns. **We would be grateful if you are willing to increase your score, and happy to discuss further**.
> >
> > Authors.

---

### Official Review · Reviewer_WdLV · 2024-11-03

**Soundness:** 2
**Presentation:** 2
**Contribution:** 2
**Rating:** 3
**Confidence:** 4

**Summary:**

The paper introduces Chemma-RC, a multimodal large language model designed for recommending chemical reaction conditions. It integrates SMILES, reaction graphs, and textual descriptions to improve prediction accuracy, supported by a dataset of 1.2 million question-and-answer pairs. Experimental evaluations on benchmark datasets demonstrate its effectiveness in various contexts. The Perceiver module enhances alignment between data modalities, contributing to the model's strong performance.

**Strengths:**

Chemma-RC captures the complexity of chemical reactions more comprehensively by combining multiple modalities, including SMILES, graph structures, and textual corpora. This integration may enhance the model's understanding and predictive capabilities regarding reaction conditions. Chemma-RC is trained on a dataset of 1.2 million question-and-answer pairs, significantly enhancing the model's learning efficacy and accuracy, especially in high-throughput experiments. By incorporating text-augmented instruction prompts, Chemma-RC effectively guides the model in learning and generating reaction conditions.

**Weaknesses:**

Integrating multiple modalities enhances the model's abilities but also complicates it, potentially leading to longer training durations and increased debugging challenges.
The model depends on extensive, high-quality multimodal data for training, which may be difficult to obtain, particularly in data-scarce environments.
The complexity of the model can hinder its interpretability, making it challenging to trace the rationale behind specific predictions, a crucial aspect in chemistry.
The model's high complexity may result in overfitting, especially when training datasets are small or noisy.
Although effective with existing multimodal data, the model might struggle to adapt to new data types or domains without adequate training examples.
The model’s success is closely tied to the quality of its training data; poor or biased data can lead to inaccurate reaction condition predictions.
Developing and training such a sophisticated model may require significant computational power and time, which could be a challenge for some research groups.
The model's performance may differ across various chemical conditions or experimental contexts, restricting its overall transferability.

**Questions:**

In Section 4.4, the ablation study highlights performance variations across different data types (SMILES, Graph, Corpus). However, it would be helpful to specify how the model utilizes each modality to enhance predictions.
In Section 4.6, the term "top-1 partial match accuracy" in the results table requires clarification. The distinction between partial and complete matches, especially regarding reagent and solvent predictions, is not clearly defined.

---

> ### Author Response · Authors · 2024-11-21
> **Response to Weakness from Reviewer WdLV**
>
> #### Dear reviewer WdLV,
> #### Thank you for your valuable feedback. Your comments are very impartial and reasonable.  We identify several of your concerns based on your comments and provide detailed responses to address them step by step. We hope the response will make you satisfied.
>
> * #### Data obtain and transferability
> #### We agree with you that Chemma-RC depends on extensive, high-quality multimodal data for training. However, we also illustrate the transferability of our models by few-shot and zero-shot evaluation on two different HTE datasets. Firstly,  we evaluate the zero-shot condition recommendation capabilities of Chemma-RC on High-Throughput Experimentation (HTE) datasets (Section 4.6). Chemma-RC recommends a ligand under a pre-defined solvent-base combination of conditions. As shown in Figure. 3, we randomly select six cases for performance evaluation. The referenced bases, solvents, and ligands can be found in the reaction formula, which has been annotated by 'B', 'S', and 'L'. In Figure. 3, under the combination of CsOAc and DMAc, Chemma-RC identifies the XPhos ligand, which results in a higher yield. For 15 of the 16 base-solvent combinations, the recommended ligand performs best in terms of the median value of reaction yields, suggesting that Chemma-RC can recommend ligands with higher yields.
>
> #### Additionally, we evaluate the performance of Chemma-RC for the **reaction reactivity prediction** task using few-shot learning, highlighting its transferability and effectiveness in data-scarce domains. We select the Ni‐catalyzed Suzuki‐Miyaura Couplings reaction for evaluation. This HTE dataset consists of 480 reaction entries across five template reaction types. Reactants, reaction conditions, and products are utilized as inputs to construct an instruction-based Q&A dataset, with the question template designed to predict reaction reactivity by GPT-4. To facilitate few-shot transfer learning, the training data we use are aligned with the baseline method in [1]. Fine-tuning with only 90 reaction entries, the model demonstrates the following performance. The table below reports the accuracy of reaction reactivity prediction.
>
> |Reactions | Chemma-RC| Baseline method|
> | ----- | ------ | ------ |
> | Raection 1 | 0.86 | 0.84 |
> | Raection 2 | 0.82 | 0.88 |
> | Raection 3 | 0.87 | 0.76 |
> | Raection 4 | 0.93 | 0.79 |
> | Raection 5 | 0.83 | 0.86 |
>
> #### From the results, we can see that Chemma-RC demonstrates higher accuracy in several reactions compared to the Multi-Threshold method. Particularly noteworthy is **Reaction 3**, where Chemma-RC achieves an accuracy of **0.87**, substantially outperforming the Multi-Threshold approach at 0.76. These results highlight Chemma-RC's strong generalization performance in reaction reactivity prediction and its potential applicability in data-scarce domains, such as HTE datasets.
> #### [1] LeSueur A, Tao N, Doyle A, et al. Multi‐Threshold Analysis for Chemical Space Mapping of Ni‐Catalyzed Suzuki‐Miyaura Couplings[J]. European Journal of Organic Chemistry, 2024: e202400428.
>
>
>  (to be continued)

---

> ### Author Response · Authors · 2024-11-21
> **[Continued] Response to Weakness from Reviewer WdLV**
>
> #### Dear reviewer WdLV,
> #### Thanks for your comments again! We continue to response to your concerns step by step. Here, we focus on discussing the complexity and computational power of our model.
> * ####  Complexity of the model
> #### We agree with you that the model's high complexity may result in overfitting, especially when training datasets are small or noisy. Thus, we design several strategies to help debug and optimize our training.
> ####  1. Regularization Techniques:
> #### &ensp; We incorporate dropout layers at various stages of the model to prevent overfitting by randomly deactivating a fraction of neurons during training.
> #### &ensp; We use warm-up strategies for the decay of the learning rate. The learning rate is set to 9.65e-6 and warm-up step is set to 5000.
> #### 2. Validation Strategies:
> #### &ensp; **Cross-Validation**: We employ k-fold cross-validation to evaluate the model's performance across different subsets of the data, ensuring that our results are consistent and not tied to a specific data split.
> #### &ensp;  **Independent Test Set**: The model is tested on an independent, unseen dataset to assess its performance in real-world conditions. This approach provides a clear measure of the model's generalization capabilities.
> ####  3. Monitoring Mechanisms:
> #### Early stopping criteria are implemented to halt training once the model's performance on a validation set ceases to improve, preventing unnecessary training that could lead to overfitting.
>
> * #### Computational power and time
>
> #### Thanks for your concern about the computational resource for training Chemma-RC. As we said before, we only update **0.3** billion parameters out of the total **7** billion parameters. This makes the training more efficient. Also, the training process is conducted with a batch size of 16 for fewer than 6 epochs over 48 hours, utilizing a GPU configuration of 2×48 GB NVIDIA A6000 GPUs. The inference process is highly efficient and can be performed using a single 80 GB NVIDIA A800 GPU.
>
> #### Here are all responses to weaknesses! Let us know if anything is missing or confusing. We are happy to further discuss and keep improving our manuscript!

---

> ### Author Response · Authors · 2024-11-21
> **Response to Questions from Reviewer  WdLV**
>
> #### Dear reviewer WdLV,
> ####  Thank you very much for the helpful suggestion and question. We want to address them correspondingly as follows:
>
> #### **(for modality integration)**
> #### We explain how the SMILES, reaction graphs, and textual corpus are integrated into this unified representation. To be specific, given a reaction, we retrieve a relevant corpus—a paragraph containing contextual information that closely resembles the reaction—and populate the <Corpus> placeholder with this data. Next, the reaction is converted into its corresponding SMILES representation, which is then inserted into the <Reaction SMILES> placeholder. Finally, we introduce two additional placeholders, <Reaction> and <Graph>, designed to accommodate the reaction and graph-based representations, respectively. In instruction fine-tuning, all reaction embedding representations are extracted by reaction encoders. Via the modality alignment module, all embeddings are inserted into token placeholders to align text-related tokens in language space. We also give pseudo-code as follows to explain this integration process.
>
> ```
> # Key part 1: map transformer-based reaction feature
> word_embed = self.word_proj(word_embed)
> word_embed = word_embed.repeat(react_embed.size()[0], 1, 1)
> react_embed = torch.cat([react_embed, word_embed], dim=1)
> smiles_react_tokens = linear_layer(react_embed) # to make 128 tokens
>
> # Key part 2: map graph-based reaction features
> graph_embed = self.word_proj(graph_embed)
> graph_react_tokens = linear_layer(graph_embed) # to make 3 tokens
>
> # Key part 3:
> reaction_tokens = torch.cat([smiles_react_tokens, graph_react_tokens], dim=1)
>
> # Key part 4: modality projection
> reaction_tokens_from_smiles = self.perceiver_proj_smiles(smiles_react_tokens)
> reaction_tokens_from_graphs = self.perceiver_proj_graphs(graph_react_tokens)
>
> # concat token
> final_token = torch.cat([reaction_tokens_from_smiles, reaction_tokens_from_graphs, text_q], dim=1)
> ```
>
> #### **(for metric)**
> #### Thanks for your attention. We mentioned "top-1 partial match accuracy" in Section 4.5 Generalization Performance. (: Maybe you refer to this part. To evaluate the generalization capabilities of Chemma-RC, we integrate the two split condition labels with dot separators as the condition ground truth for reagents and solvents prediction, respectively. Then we compare the condition predictions with the ground truth and adopt the metric of partial matched accuracy. Different from the complete matched accuracy that requires perfect matching between predictions and labels, the partial matched accuracy is more suitable to test the generalization capacity, which focuses more on whether the predicted results match a substitutable part of the ground truth. For example, if the predicted result is ‘[Na+].[OH-]’ and the condition label is ‘CO.[Na+].[OH-]’, we consider that the prediction partially matches the ground truth, but not completely. From the evaluation results of the partial matched accuracy, we can conclude that our Chemma-RC can successfully distinguish reagents from the combination of all conditions in a reaction, as it learns the relationships between reaction conditions effectively.
>
> #### Here are all our responses. Let us know if anything is missing or confusing. We are happy to further discuss and keep improving our manuscript!

---

### Author Response · Authors · 2024-11-22
**General response and revision update notice**

#### Dear Reviewers,
#### We are grateful for the valuable feedback and constructive suggestions provided during the review process. We have revised the paper accordingly and would like to highlight the key areas where changes have been made. All revisions in the manuscript are marked in red for your convenience. We sincerely hope these improvements address your concerns and contribute to the overall quality of the paper. Please feel free to reach out if there are any new questions or points for further discussion.
#### **Data**
* #### We have provided a detailed illustration of the construction process for the instruction Q&A datasets. This includes an in-depth explanation of methodologies and frameworks designed, ensuring that the dataset's foundation is clear and reproducible. (3FKb Q5, GYuz Q1, od79 W4)
#### **Experiments**
* #### We have included baseline models for comparison purposes, such as nach0, to provide a more robust context for our findings. (GYuz Q3,
* #### The performance of Chemma-RC has been thoroughly evaluated under different dataset splitting strategies, including both random split (RS) and time-based split (TS). (3FKb Q1)
* #### To ensure comprehensive analysis, we performed ablation studies focusing on models trained solely on mono modality (graph-based) as well as those trained on cross modalities (graph-based features and SMILES representations). Further, we add more discussions and explanations for the contributions of each modality. (WdLV Q1, 3FKb Q7, GYuz Q2, od79 W3)
* #### We also conducted a reaction reactivity prediction task using few-shot learning to assess the model's adaptability and effectiveness in limited-data scenarios.
* #### Detailed training settings and computational requirements have been added to provide transparency and reproducibility. (od79 Q6, 3FKb W3,4,6)
#### **Model Configuration**
* #### Additional details regarding the model's efficiency and debugging challenges have been incorporated to outline the developmental hurdles and solutions. （3FKb Q9, od79 W4)
#### **Metrics**
* #### We have provided detailed explanations regarding 'top-1 partial match accuracy' and clarified the distinction between partial and complete matches to ensure that the evaluation metrics are well understood. (WdLV Q2, 3FKb Q6, 3FKb Q10)

#### Thank you once again for your thorough and insightful review. We look forward to any further feedback and are eager to discuss any new aspects or concerns.
#### Sincerely,
#### The Authors

---

### Note · Authors · 2025-08-03

I have read and agree with the venue's withdrawal policy on behalf of myself and my co-authors.

---

### Meta-Review · Area_Chair_k6mF · 2024-12-20

**Metareview:**

The work introduces a multimodal LLM-based model for predicting chemical reaction conditions and yield.

Two reviewers voted for accepting the work, while two other voted for rejection. Among the strengths, reviewers (WdLV and 3FKb) noted the extensive instruction tuning dataset, very encouraging (as noted by GYuz, though it is important to note it was demonstrated on one dataset) OOD results, and contributing to the important field of using LLMs in the context of chemical reactivity (GYuz).

Among weaknesses, reviewers noted overall complexity of the solution (WdLV, 3FKb, GYuz), and asked for additional baselines (3FKb). During the rebuttal phase, Authors have added requested baselines such as nach0, with the proposed models achieving significant gains.

Another significant issue, though not raised specifically by any reviewer, is novelty. The paper is technically advanced and nontrivial to execute, but conceptually rather straightforward application of a large language model with fused (using a Perceiver) modality information. As such it is not the best target for ICLR, and is better suited for a more specialized and application oriented venue. Relatedly, the paper shows small improvements in Table 2 while large improvements in Table 5 (AC is not sure what is the difference between Table 2 and Table 5 evaluation setting) compared to a closely related TextReact, and is not compared to TextReact on USPTO 500MT Condition. From these results it is not yet possible to fully understand the improvement on top of TextReact. This issue is further excarberated by the fact that (to the best knowledge of AC) the ablations are not done by removing modality at train time but merely at test time.

To summarize, despite the strengths of the paper, its limited novelty and complexity rank the paper slightly below the bar for acceptance. I hope the comments will be helpful in improving your work and thank you for submitting to ICLR.

**Additional Comments On Reviewer Discussion:**

During the rebuttal, reviewers raised concerns about baseline comparisons, novelty, and model complexity, which were addressed through additional baselines, clarifications on results, and detailed responses. The reviewers’ insights were weighed, and while the improved baselines demonstrated the model’s strength, the paper's limited conceptual novelty and execution complexity influenced the final decision.

---

### Decision · Program_Chairs · 2025-01-22

Reject